# Mixed layer depth dominates over upwelling in regulating the seasonality of ecosystem functioning in the Peruvian Upwelling System

Tianfei Xue[1], Ivy Frenger[1], A.E. Friederike Prowe[1], Yonss Saranga José[1], and Andreas Oschlies[1,2]

[1]GEOMAR Helmholtz Centre for Ocean Research Kiel, Kiel, Germany
[2]Christian-Albrechts-University Kiel, Germany

**Correspondence:** Tianfei Xue (txue@geomar.de)

**Abstract.** The Peruvian Upwelling System hosts a marine ecosystem with extremely high productivity. Observations show that the Peruvian Upwelling System is the only Eastern Boundary Upwelling System (EBUS) with an out-of-phase relationship between seasonal surface chlorophyll concentrations and upwelling intensity. This "seasonal paradox" triggers the following questions: (1) What are the unique characteristics of the Peruvian Upwelling System, compared with other EBUS, that lead to the out-of-phase relationship; and (2) How does the seasonal paradox influence ecosystem functioning? Using observational climatologies for four EBUS, we diagnose that the Peruvian Upwelling System is the only one to reveal that intense upwelling coincides with deep mixed layers. We then apply a coupled regional ocean circulation-biogeochemical model (CROCO-BioEBUS) to assess how the interplay between mixed layers and upwelling regulates the seasonality of surface chlorophyll in the Peruvian Upwelling System. Our model reproduces the "seasonal paradox" within 200 km off the Peruvian coast. We confirm previous findings regarding the main contribution of mixed layer depth to the seasonality of chlorophyll, relative to upwelling. Deep mixed layers in austral winter cause vertical dilution of phytoplankton and strong light limitation, impacting growth. The effect of advection, though second-order, is consistent with previous findings for the Peruvian system and other EBUS, with enhanced offshore export opposing the coastal buildup of biomass. In addition, we find that the relatively colder temperatures of upwelled waters slightly dampen phytoplankton productivity and further slow the buildup of phytoplankton biomass. This impact from the combination of deep mixed layers and upwelling propagates through the ecosystem, from primary production to export and export efficiency. Our findings emphasise the crucial role of the interplay between mixed layer depth and upwelling and suggest that surface chlorophyll may increase, along with a weakened seasonal paradox, in response to shoaling mixed layers under climate change.

# 1 Introduction

The Peruvian Upwelling System (PUS) hosts a disproportionally productive ecosystem, supporting 10% of the world's fishing yield while covering only 0.1% of the ocean area (Chavez et al., 2008). As one of the Eastern Boundary Upwelling Systems (EBUS), winds favouring upwelling raise cool, nutrient-rich waters to the surface, supporting high primary production and fish yield. Simultaneously, high primary production, together with subsequent export and remineralisation contributes to the formation of a sub-surface oxygen-deficient zone which is particularly shallow and intense in the PUS (Fuenzalida et al., 2009; Stramma et al., 2010; Getzlaff et al., 2016). Particularly due to its high productivity, the response of the PUS to climate change is of great social and economic interest (Pauly et al., 1998; Bakun, 1990; Bakun et al., 2010), and a variety of studies have investigated how physical and biogeochemical processes influence the production of phytoplankton as well as its potential links to ecosystem functioning in the PUS.

While the PUS has been frequently compared to other EBUS (e.g., the Benguela, California, and Canary Systems), it is set apart by how surface chlorophyll responds to the variation of upwelling on a seasonal scale. The high productivity of EBUS primarily benefits from the upwelling of nutrient-rich waters, driven by alongshore equatorward winds. Hence, it is commonly assumed that the magnitude of phytoplankton biomass in EBUS is directly correlated with the wind-driven upwelling intensity (Bakun, 1973). However, in the PUS, upwelling intensity and surface chlorophyll are not correlated on a seasonal scale (hereafter referred to as "seasonal paradox"; Chavez, 1995; Thomas et al., 2001; Echevin et al., 2008; Chavez and Messié, 2009). Instead, they are out of phase, with the lowest surface chlorophyll concentration in austral winter corresponding to maximum upwelling intensity (Calienes et al., 1985). Echevin et al. (2008) used a regional coupled physical-biogeochemical model to simulate the "seasonal paradox" and found that deep mixed layers caused dilution of surface phytoplankton, reduced growth due to limited light, and subsequently reduced iron levels (as phytoplankton require more iron under low-light conditions). This ultimately leads to low chlorophyll under strong upwelling conditions. Results from Messié and Chavez (2015) corroborated iron and light limitations found in Echevin et al. (2008), showing additionally that relatively strong offshore advection in austral winter regulated the buildup of phytoplankton and thus also contributed to the seasonal paradox. Guillen and Calienes (1981) suggested that lower surface irradiation in winter might amplify light limitation and further limit phytoplankton growth, while insolation was found not to play a major role in Echevin et al. (2008). Additionally, Echevin et al. (2008) concluded that temperature played no role in regulating phytoplankton growth. Despite previous research on surface chlorophyll seasonality, uncertainty still remains regarding why the seasonal paradox occurs only in the PUS and not in the other EBUS, and it is unexplored how the seasonal paradox affects ecosystem functioning.

This study addresses the following key questions: (1) what are the unique characteristics of the PUS, compared to other EBUS, that lead to this seasonal paradox; (2) what are the mechanisms that cause low surface phytoplankton in winter; and (3) how do these mechanisms affect ecosystem functioning.

## 2 Data and Methods

### 2.1 Regional ocean circulation-biogeochemical model: setup and simulation

We use a climatological simulation of the three-dimensional regional ocean circulation model CROCO (Coastal and Regional Ocean COmmunity model; Debreu et al., 2012) coupled with the biogeochemical model BioEBUS (Biogeochemical model for the Eastern Boundary Upwelling Systems; Gutknecht et al., 2013) for this study.

The same technical setup, including the model grid, is used as in José et al. (2017), along with an updated version of the ocean circulation model CROCO. CROCO is the next generation of the ROMS AGRIF model (Tedesco et al., 2019), and is a free-surface and split-explicit regional ocean model system (ROMS; Shchepetkin and McWilliams, 2005). We employ a two-way nesting approach, with the larger coarser-resolution domain covering the Southeast Pacific and the smaller higher-resolution domain focusing on the PUS. The larger domain has a 1/4° resolution, spanning from 69°W to 120°W and from 18°N to 40°S. The embedded "child" domain has a resolution of 1/12° and extends from 5°N to 31°S and from 69°W to 102°W (Fig. 1a-b & A1) and is used in this study. Both the coarse- and fine-resolution domains use 32 sigma levels in the vertical direction, with finer resolution towards the surface and shallower regions. The surface layer thickness ranges from 0.5 m in the coastal region (water depth around 50 m) to around 3 m in the offshore region (water depth of more than 4000 m). Initial and boundary conditions are provided by the monthly climatological SODA reanalysis (Simple Ocean Data Assimilation; Carton and Giese, 2008) from 1990−2010. Surface forcing is based on the monthly climatological heat and freshwater fluxes from COADS (Comprehensive Ocean-Atmosphere Data Set; Worley et al., 2005), along with wind data from QuikSCAT (Quick Scatterometer; Liu et al., 1998). The physical setup is the same as in José et al. (2017) and has been evaluated therein, showing that the model reproduces the circulation of the region reasonably well.

The biogeochemical BioEBUS model used in this study was developed explicitly for applications to EBUS and oxygen minimum zones (Gutknecht et al., 2013). BioEBUS is a nitrogen-based model, originating from the $N_2P_2Z_2D_2$ model by Koné et al. (2005). It simulates two phytoplankton and two zooplankton groups: small and large phytoplankton, along with micro- and mesozooplankton. Furthermore, there are two detritus pools, categorised by size. BioEBUS resolves the N species (nitrate, nitrite and ammonium) and simulates processes under oxic, hypoxic and suboxic conditions (e.g., remineralisation, nitrification, denitrification and anammox). The BioEBUS model was first used to study the Peruvian marine biogeochemistry by Montes et al. (2014), and is capable of producing a realistic simulation of the oxygen distribution. Initial and boundary conditions for nitrate and oxygen are taken from CARS (CSIRO (Commonwealth Scientific and Industrial Research Organisation) Atlas of Regional Seas; Ridgway et al., 2002), and initial conditions for phytoplankton are based on monthly climatological SeaWiFS (Sea-viewing Wide Field-of-view Sensor; O'Reilly et al., 1998) estimates. A detailed description of these biogeochemical processes can be found in Gutknecht et al. (2013). The parameter settings are the same as in José et al. (2017), except for a few adjustments of biological parameters (Table. A1) to improve the fit between the simulated ecology, in particular phytoplankton and zooplankton, and the observations.

CROCO-BioEBUS is run in coupled mode from the beginning of the simulation. The time-stepping of the physical model is the same as the coupling time step, with a duration of 1200 seconds. The time-stepping of the biogeochemical model has a

duration of 400 seconds. The coupled model is run for a 25-year spin-up period. Physical and biogeochemical fields are spun up after one year for the upper 10 m, while waters in the depth range of upwelling (100 m) require 3–10 years longer to reach a statistical quasi-equilibrium (Fig. C1). We run the model for a total of 30 climatologically-forced years, using the last five years for the analyses. As we observe from the surface ecology, our results are not sensitive to deep ocean spin-up. This study focuses on the 200 km band off the Peruvian coast (white line region in Fig. 1a-b), which shows clear seasonal variation as well as strong upwelling.

## 2.2 Analysis approaches

To assess the seasonal variance of phytoplankton biomass concentration in each grid box ($C$), we analysed the budget of the phytoplankton biomass and how its tendency is driven by physical versus biological processes:

$$\frac{\partial C}{\partial t} = PHY(C) + BIO(C) \tag{1}$$

with $[BIO = PP - GRAZ - MORT - EXU - SINK; \; PHY = MIX + ADV]$

PHY represents the physical processes, including advection $ADV$ and mixing $MIX$, whereas BIO represents the biological processes, namely primary production $PP$, consumptive mortality $GRAZ$, natural mortality $MORT$, exudation $EXU$ and sinking $SINK$. All biological and physical fluxes were saved monthly from the simulation, with units of mmol N m$^{-3}$ s$^{-1}$, and we integrated the terms offline over the mixed layer depth (MLD) using the croco-tools provided for post-processing (https://www.croco-ocean.org/download/croco-project/). The mixing term also includes entrainment from varying MLD as a minor contribution.

We analysed in detail the drivers of $PP$, which was calculated online by multiplying phytoplankton concentration ($C$) and the growth factors ($L_{(PAR)}, L_{(T)}, L_{(N)}$)

$$PP = C \cdot L_{(PAR)} \cdot L_{(T)} \cdot L_{(N)} \tag{2}$$

where $L_{(PAR)}, L_{(T)}, L_{(N)}$ represent the light-, temperature- or nitrogen-related growth factors, respectively. Here, the phytoplankton growth rate was defined as a multiplicative function of the light-, temperature- or nitrogen-related growth factors. The limitation experienced by phytoplankton within the mixed layer $L_{\mathrm{mld}}$ is calculated offline from each growth factor ($L_{(PAR)}$, $L_{(T)}$ and $L_{(N)}$), using phytoplankton concentration ($C$) within the mixed layer as a weight (Eq. 3). Light-, temperature- and nitrogen-related growth factors that each phytoplankton cell experienced were computed online.

$$L_{\mathrm{mld}} = \frac{\sum_0^{\mathrm{mld}} L_{(PAR)} \cdot L_{(T)} \cdot L_{(N)} \cdot C}{\sum_0^{\mathrm{mld}} C} \tag{3}$$

For the analysis, we attributed the seasonal change of the average phytoplankton biomass concentration ($\Delta C_{\mathrm{mld}}$) within the mixed layer to the change in the integrated phytoplankton content within the mixed layer ($\Delta B_{\mathrm{mld}}$), as well as the change in volume of the mixed layer ($\Delta V_{\mathrm{mld}}$). Using the chain rule and the condition that $V^2 >> V\Delta V$, we approximated a discrete change in the mixed layer tracer concentration ($\Delta C_{\mathrm{mld}}$) as follows:

$$\Delta C_{\mathrm{mld}} = \frac{1}{V_{\mathrm{mld}}} \Delta B_{\mathrm{mld}} - B_{\mathrm{mld}} \frac{\Delta V_{\mathrm{mld}}}{V_{\mathrm{mld}}^2} = \frac{B_{\mathrm{mld}}}{V_{\mathrm{mld}}} \frac{\Delta B_{\mathrm{mld}}}{B_{\mathrm{mld}}} - \frac{B_{\mathrm{mld}}}{V_{\mathrm{mld}}} \frac{\Delta V_{\mathrm{mld}}}{V_{\mathrm{mld}}} \tag{4}$$

To assess the relative contributions, we then divided by $C_{\mathrm{mld}} = B_{\mathrm{mld}} \cdot V_{\mathrm{mld}}^{-1}$ to obtain

$$\frac{\Delta C_{\mathrm{mld}}}{C_{\mathrm{mld}}} = \frac{\Delta B_{\mathrm{mld}}}{B_{\mathrm{mld}}} - \frac{\Delta V_{\mathrm{mld}}}{V_{\mathrm{mld}}} \tag{5}$$

which allowed us to attribute decreased concentration of phytoplankton in the mixed layer $C_{\mathrm{mld}}$ to a decrease in the phyto-
plankton biomass $B_{\mathrm{mld}}$ or an increase in the mixed-layer volume $V_{\mathrm{mld}}$, and vice versa.

## 2.3 Observational data and model assessment

For EBUS comparisons, we digitised SeaWIFS climatological surface chlorophyll and upwelling (a combination of Ekman
transport and Ekman pumping; Messié et al., 2009), estimated based on winds from QuikSCAT, from Chavez and Messié
(2009). Additionally, we used surface nitrate data from the World Ocean Atlas (WOA; Garcia et al., 2019), the gridded ARGO
mixed layer dataset (http://mixedlayer.ucsd.edu/; Holte et al., 2017), monthly climatologies of MODIS sea surface temperature
(SST) and chlorophyll data (https://oceancolor.gsfc.nasa.gov/data/aqua/) to analyse and evaluate the model results.

The model was evaluated based on averages over the focus region, with monthly observational data. The correlation co-
efficient between the model simulation and observations, the root mean square error (RMSE) and the normalised standard
deviation (SD) of the observations relative to the model results are shown in a Taylor diagram as a summary of the evalua-
tion (Fig. 1c; Taylor et al., 1991, a comparison of the spatial pattern and the seasonal cycles of variables is provided in the
appendix, see Figs. B1-B4). Model results fit the observational data reasonably well. The model effectively simulated SST
with $R$>0.95, 1<$\sigma^*$<1.2 and $RMSE^*$<0.4 ($R$: correlation coefficient, $\sigma^*$: normalised SD, and RMSE$^*$: normalised RMSE).
It also captured the observed seasonal cycle well, though it produced slightly stronger seasonal variations compared to those
from the observational data. Although the seasonal variation was somewhat overestimated, the simulated MLD (defined based
on a 0.2 $^oC$ temperature difference criterion) remained largely within the observed range of ARGO-based MLD (Fig. B3). As
for biogeochemical variables, the model effectively simulated surface nitrate, with $R$>0.95, 0.6<$\sigma^*$<1 and $RMSE^*$<0.4, but
overestimated the nitrate compared to WOA. However, cruise data (Fig. B2c-d) show that the overestimation could have arisen
from WOA failing to capture the high-surface nitrate concentration in the coastal region under strong upwelling. A comparison
of the simulated and observed seasonal cycle of surface chlorophyll in the focus region (Fig. 1d) revealed that modelled chloro-
phyll generally followed the seasonal trend of satellite and *in situ* data, with the amplitude of the seasonal cycle in between
amplitudes from satellite and *in situ* data. Overall, the model showed reasonably good agreement with observational data on a
seasonal scale, sufficiently supporting an investigation of the seasonal paradox with CROCO-BioEBUS.

## 3 Results

### 3.1 Anticorrelation of chlorophyll and upwelling: The seasonal paradox only appears in the Peruvian upwelling system

Compared with other EBUS (spatial extent of EBUS regions indicated in Fig. A2), the Peruvian system is unique in that
it shows a clear anticorrelation between surface chlorophyll concentration and upwelling intensity on a seasonal scale, with

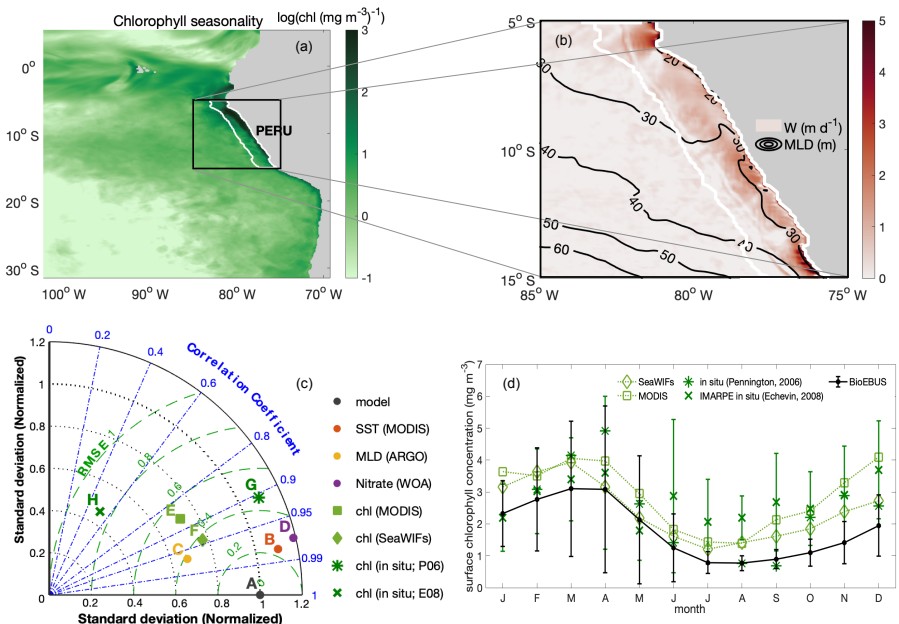

**Figure 1.** (a) Spatial distribution of the amplitude of the annual cycle of surface chlorophyll in log scale ($\log(\mathrm{chl}\,(\mathrm{mg\,m}^{-3})^{-1})$); (b) Map of annual mean upwelling velocity $w\,(\mathrm{m\,d}^{-1})$ at the bottom of the mixed layer, with contour lines indicating MLD (m). White lines highlight the focus area; (c) Taylor diagram for seasonal SST (red), MLD (yellow), surface nitrate (purple) and chlorophyll (green) concentrations. The black dot indicates the model simulation as a reference. The radial distance from the origin is proportional to the standard deviation, normalised by the standard deviation of the data. The green dashed lines show the RMSE. The correlations between model and observations are given by the azimuthal position; (d) Seasonal cycles of surface chlorophyll concentration from model simulation (black solid line), satellite data (dotted line; SeaWIFS (diamond) and MODIS (square)) and in-situ data (digitised from Pennington et al. (2006, star, 250 km band off the coast) and Echevin et al. (2008, cross)). Error bars indicate the standard deviation.

lowest chlorophyll concentrations when upwelling is most intense (Fig. 2a, $R^2 = 0.71$; Chavez and Messié, 2009). While the surface chlorophyll in the Benguela system does not feature a strong seasonality, surface chlorophyll closely follows upwelling intensity in the California ($R^2 = 0.92$) and Canary ($R^2 = 0.88$) systems, suggesting that upwelling of nutrient-rich waters fuels the increase in chlorophyll. Indeed, comparatively low surface nitrate concentrations indicate that nitrate is depleted, potentially limiting phytoplankton growth throughout the year in the California system and for approximately half the year in the Canary system (Fig. 2b). In contrast, the Benguela ($R^2 = 0.63$) and Peruvian systems ($R^2 = 0.90$) feature replete surface nitrate over most of the year. Because higher nitrate concentrations correlate with lower chlorophyll in these cases, nitrate is not observed to be a limiting factor.

In the Peruvian system, a strong relationship exists between deepening mixed layers and decreasing chlorophyll (Fig. 2c, $R^2 = 0.91$), which supports the notion that dilution of phytoplankton over a deeper mixed layer and/or light limitation plays a

role, as found by Echevin et al. (2008). The California system shows a similar response to mixed layer variations ($R^2 = 0.62$), suggesting that the same process may play a role there as well. Additionally, surface chlorophyll shows significant correlation with SST in the Peruvian (Fig. 2d, $R^2 = 0.73$) and California systems ($R^2 = 0.65$), suggesting that increasing temperatures stimulate phytoplankton growth.

Strikingly, the Peruvian system is the only one of the four EBUS where strong upwelling coincides with deep MLD (Fig. 2e, $R^2 = 0.79$). The Canary and Benguela systems exhibit pronounced seasonality either in upwelling or in mixed layer depth, respectively. In the California system, the relationship of upwelling and mixed layer depth is opposite to that of the Peruvian system, with the strongest upwelling occurring in the shallowest mixed layers. Given the paradox that strong upwelling in the Peruvian system occurs at the time of the yearly chlorophyll minimum, it is intuitive that the concurrent deep mixed layers offset the positive impact of upwelled nutrients. Nutrient enrichment would only stimulate higher productivity if the region was nutrient limited. If concentrations are already elevated, adding more nutrients would have a weak impact. We will further investigate the interplay of the seasonality of mixed layers and upwelling in the Peruvian system in the following sections.

## 3.2 Modelled phytoplankton biomass, dissolved inorganic nitrogen, upwelling and the MLD in the Peruvian system

We used a regional ocean circulation model, coupled to a marine biogeochemical model (CROCO-BioEBUS), to further analyse the Peruvian system (see the "Data and Methods" section). The model effectively reproduced the observed estimate of the seasonal out-of-phase relationship between surface phytoplankton biomass and upwelling intensity as well as nitrate concentrations (Fig. 2, open squares). Over the course of the year, surface chlorophyll, surface nitrogen concentrations, upwelling intensity and MLD varied by 40 - 60% relative to their annual mean values. Surface phytoplankton biomass concentration reached its maximum from late austral summer to early autumn (March to April), when upwelling was relatively weak (Fig. 3a-b). During this time window, less nitrogen is available within a shallow MLD compared with the rest of the year. In austral winter (July to September), when upwelling introduces ample nitrogen into the deep mixed layer, surface phytoplankton concentration reaches a minimum.

## 3.3 Biomass dilution by the deepening mixed layer

Dilution of phytoplankton in deepening winter mixed layers is a key driver behind the seasonality of surface phytoplankton concentration. Within the research area, the MLD showed a seasonal variation with the shallowest mixed layer in austral summer, around 10 m, and the deepest mixed layer in austral winter, around 45 m. Phytoplankton were vertically well mixed within the mixed layer throughout the year (Fig. 3c). In austral winter, within the 'deep-mixing' regime, phytoplankton were evenly distributed over a relatively deep mixed layer, diluting phytoplankton biomass. Accordingly, phytoplankton biomass concentrations in the mixed layer as well as at the surface decreased. Hence, we infer that seasonal mixed layer deepening and shoaling alone is an important factor in driving phytoplankton concentrations at the ocean surface, as observed for instance from satellite images.

While dilution caused a decrease in winter surface phytoplankton biomass, it explained only part of the observed biomass decrease. The decline persisted, even though attenuated, when integrating phytoplankton over the mixed layer (Fig. 3b). The

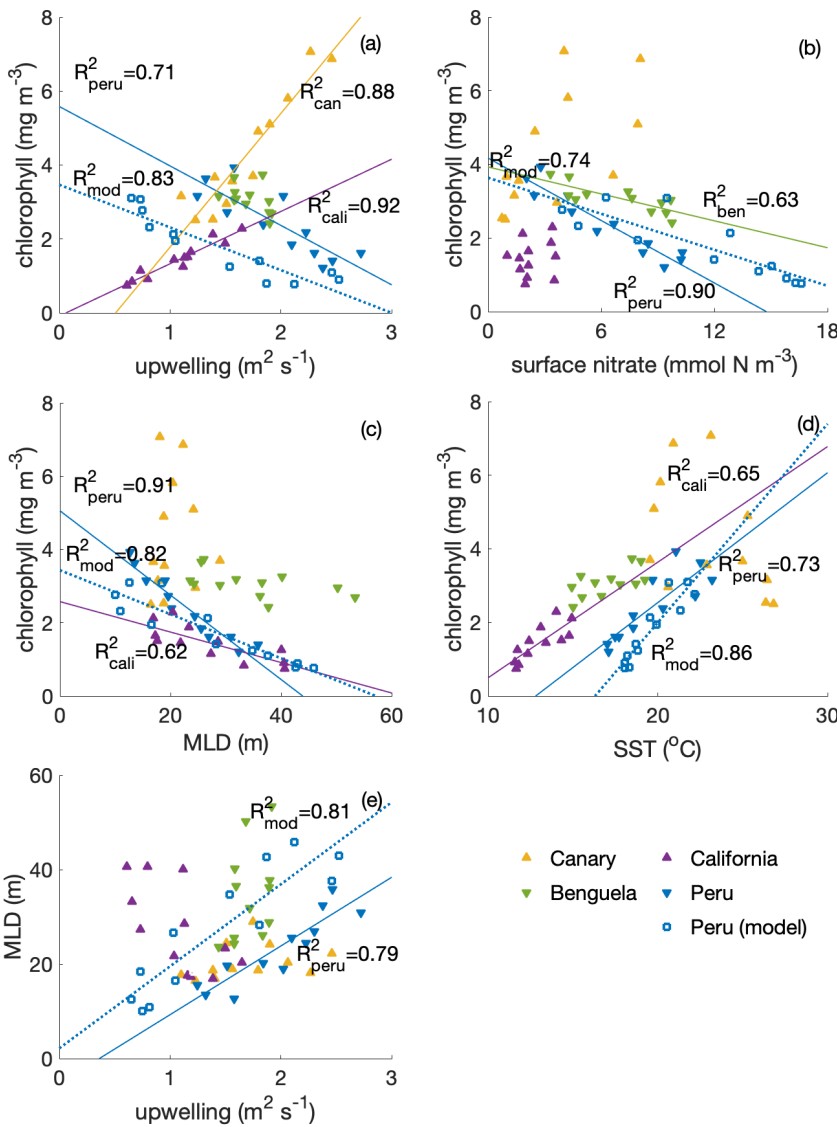

**Figure 2.** Correlations of surface chlorophyll (SeaWIFS climatology, in $\mathrm{mg\,m^{-3}}$) with (a) upwelling (a combination of Ekman transport and Ekman pumping, estimated based on winds from QuikSCAT, in Sv, digitised from Chavez and Messié (2009), for calculations see Messié et al. (2009)); (b) surface nitrate concentration (WOA, in $\mathrm{mmol\,N\,m^{-3}}$); (c) MLD (ARGO, in m); (d) SST (MODIS, in $^{\circ}$C) and (e) correlation of MLD and upwelling transport among four eastern boundary upwelling systems (EBUS). For the Peruvian system, we also show the model (CROCO-BioEBUS) results. Lines and $R^2$ values are displayed for correlations with $R^2 > 0.5$.

phytoplankton concentration at the surface and within the mixed layer declined by around 70%, while it declined by around 30% for MLD-integrated biomass between late April and late July (shaded area in Fig. 3  6, hereafter referred to as the decline

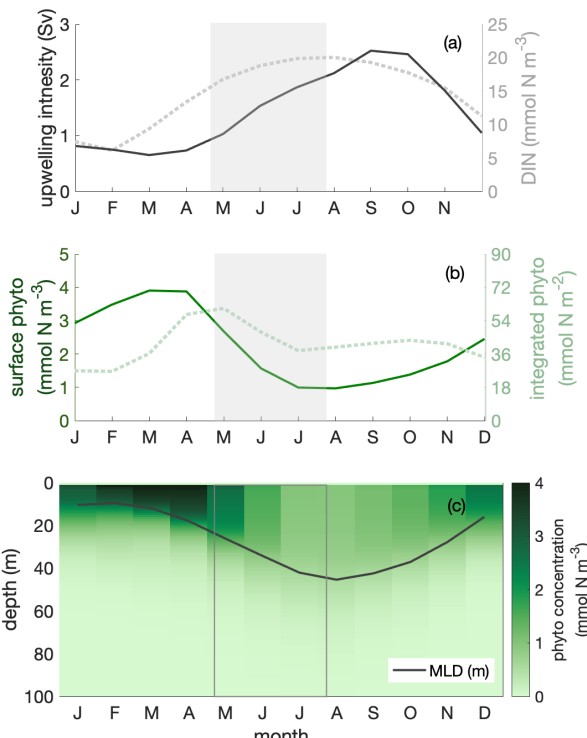

**Figure 3.** Seasonal cycles of (a) upwelling intensity (in Sv, solid line) and surface dissolved inorganic nitrogen (DIN) concentration (in $\mathrm{mmol\,N\,m^{-3}}$, dotted line); (b) surface (in $\mathrm{mmol\,N\,m^{-3}}$, solid line) and mixed layer depth (MLD)-integrated phytoplankton biomass (in $\mathrm{mmol\,N\,m^{-2}}$, dotted line); (c) phytoplankton depth-month distribution, showing the seasonal cycle of MLD (in m, solid line) within the focus region. The shaded area indicates the decline phase of MLD-integrated phytoplankton biomass.

phase). The decline of surface phytoplankton concentrations can be attributed to the decline due to the increase in mixed layer volume $\Delta V$ (dilution effect, see Eq. 5) and the decrease in biomass $\Delta B$ within the mixed layer through local biological and physical processes (see Eq. 5). During the decline phase, $\Delta V$ contributed by slightly more than half to the concentration change, while $\Delta B$ contributed slightly less than half. That is, the dilution effect due to the deepening mixed layer in the decline phase amplified the decline of surface biomass concentrations by approximately a factor of two. However, dilution could not fully explain the low phytoplankton biomass in conditions where the supply of nitrogen is ample; in such conditions, MLD-integrated biomass still declined by around 30%.

## 3.4 Biological and physical processes change the total biomass within the mixed layer

### 3.4.1 Disentangling physical and biological processes

In addition to causing dilution due to the deepening mixed layer, the imbalance of a series of biological and physical processes during the decline phase diminished phytoplankton concentrations. To disentangle their contributions to the decline of phytoplankton concentration without the complicating factor of the dilution effect, we next analysed the change of phytoplankton biomass integrated over the mixed layer (Fig. 4a) and its drivers; that is, the mixed layer budget of phytoplankton biomass (Eq. 1). We separated biological processes (primary production, grazing from zooplankton, natural mortality, exudation and sinking) and physical processes (mixing and advection) that affect the integrated biomass. Throughout the year, the net biological flux (the sum of all biological fluxes) was positive ("biological gain", Fig. 4b), thus supporting an increase in biomass. In contrast, the net physical flux, the sum of all physical fluxes, was negative ("physical loss"), therefore supporting a decrease in biomass. The time point t1 marks the seasonal maximum of the MLD-integrated phytoplankton biomass, and t2 marks the minimum at the end of the decline phase. At t1 and t2, the net biological and physical fluxes balanced (Fig. 4b-c) and the tendency of the mixed layer phytoplankton biomass was zero (Fig. 4a). Between t1 and t2 (Fig. 4b), the net biomass supply due to biological fluxes decreased more quickly than the net biomass removal due to physical fluxes, resulting in an imbalance of the fluxes and the decrease in biomass between t1 and t2.

To determine which terms from Eq. 1 mostly drove the decrease of the biomass between t1 and t2 (Fig. 5a), we integrated the change of each term over time (that is, the derivatives) between t1 and t2. Therefore, in Fig. 5b and c, if a bar was positive, the change of the term during the decline phase (t1 - t2) promoted an increase of the phytoplankton biomass, mostly as a result of reduced grazing pressure and reduced downward mixing. If the bar was negative, the change of the term during the decline phase (t1 - t2) opposed an increase in phytoplankton biomass. The "opposing terms" that acted to reduce phytoplankton biomass were the ones that contributed to the seasonal paradox; that is, decline in biomass despite increased supply of nutrients due to upwelling. These terms mostly referred to the reduced primary production, with a secondary contribution from the increased divergence due to advection. Details regarding the two major contributors, primary production and advection, are presented in the following sections.

### 3.4.2 Factors limiting primary production

Primary production changed due to variations in both the growth factor and the biomass (Eq. 2). The growth factor (calculated as in Eq. 3, Fig. 6a) combined the effects of light, temperature and nitrogen on phytoplankton growth. It showed a clear decrease of around 30% during the decline phase. Optimal phytoplankton growth conditions were reached in March, despite the low dissolved inorganic nitrogen (DIN) conditions, within the warmest and brightest environment. The lowest growth rate occurred just after the decline phase, despite relatively high nitrogen concentrations, due to limiting light and temperature conditions.

Strong light limitation experienced by phytoplankton, in combination with low temperatures, slowed growth during the decline phase. Light conditions for phytoplankton growth were optimal in March when the water was rather stratified and

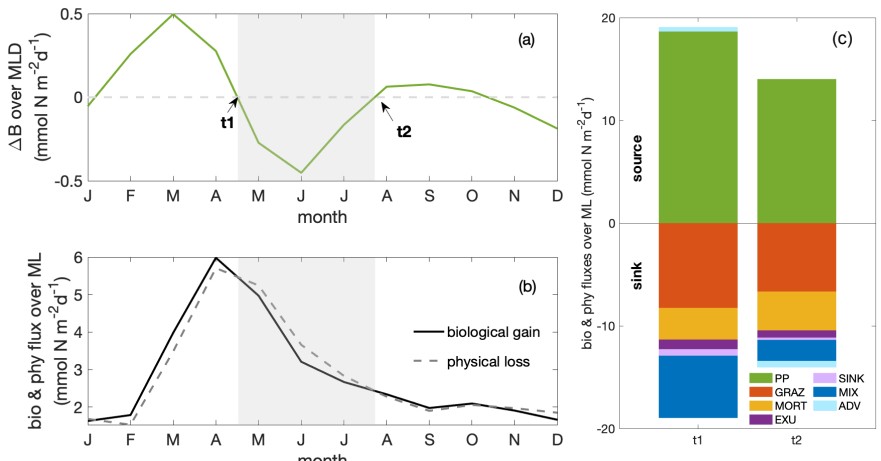

**Figure 4.** Seasonal cycles of (a) total phytoplankton biomass change ($\Delta B$; in mmol N m$^{-2}$ d$^{-1}$). Grey shading indicates the decline phase, with t1 and t2 marking the beginning and end of the decline phase; (b) phytoplankton fluxes resulting from net biological gain (bio) and net physical loss (phy, in mmol N m$^{-2}$ d$^{-1}$, as shown in Eq. 1); (c) Balancing budgets of phytoplankton fluxes at t1 and t2 (PP: primary production; GRAZ: consumptive mortality; MORT: natural mortality; EXU: exudation; SINK: sinking; MIX: mixing; ADV: advection). Fluxes are distinguished as sources (positive values) and sinks (negative values) of phytoplankton biomass, with the sum of all source and sink terms balancing at times t1 and t2. All fluxes are integrated over the MLD.

worsened over the decline phase to a minimum in August when the water column was most deeply mixed (Fig. 6a). The light-related growth factor declined by 17% during the decline phase and would decrease the growth factor by approximately
60% in the absence of other limiting factors (estimated from the product rule for differentiation and the multiplicative relation of growth factors shown in Eq. 2 & 3). Decreasing temperature was the second most important contributor in slowing the growth rate during the decline phase. The temperature-related growth factor reached its maximum by March, similar to the light-related growth factor, and reached its minimum by October. The temperature-related growth factor declined by 12% and would decrease the growth factor by around 40% during the decline phase, in the absence of other limiting factors. In
contrast, the seasonality of the growth factor due to nitrogen showed the opposite seasonality compared to the total growth factor. Clearly, light and temperature regulated primary production and overrode the effect of enhanced nitrogen supply during the decline phase. Therefore, while light was the dominant mechanism that reduced productivity towards winter, we found that temperature played a relevant secondary role.

Stronger light and temperature limitation during the decline phase were due to deeper mixing and stronger upwelling of cold
waters, respectively (Fig. 6b-c). While upwelling intensity was approximately correlated with MLD, the maximum upwelling occurred just after the deepest mixed layers. The variation of MLD-averaged light limitation was correlated ($R^2 = 0.92$) with the change of MLD. As phytoplankton were evenly distributed within the mixed layer, deeper MLD indicated that, on average, more phytoplankton were exposed to lower light conditions during the decline phase, with a minimum in August when mixed

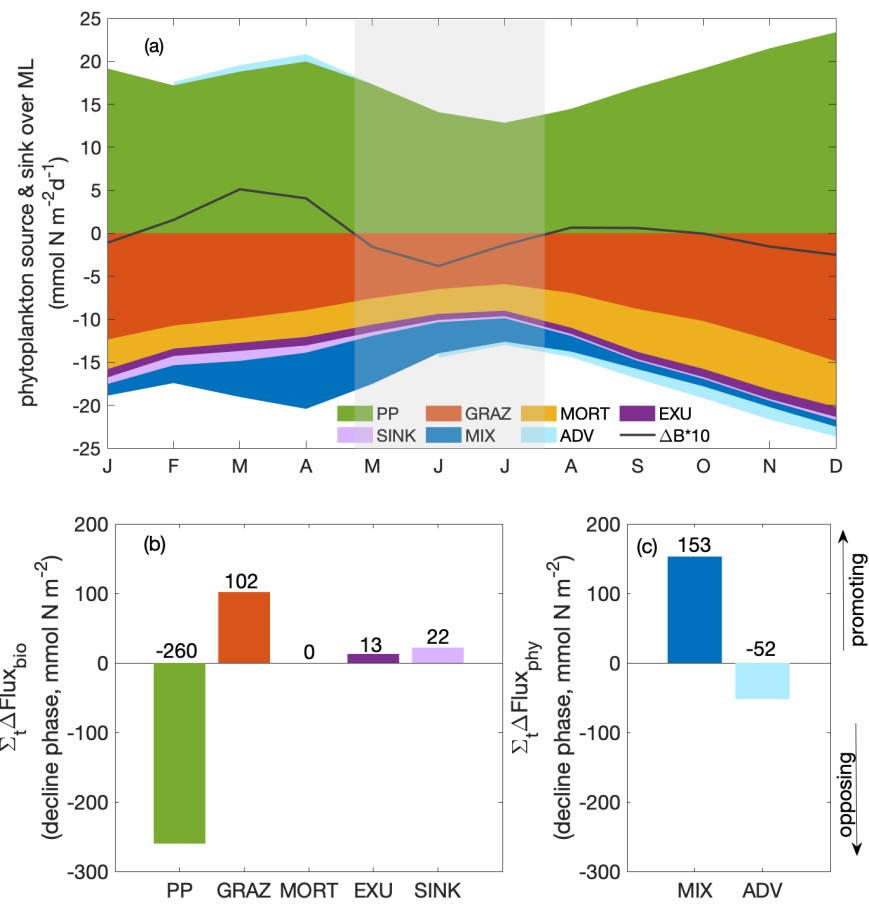

**Figure 5.** (a) Seasonal cycle of phytoplankton source and sink processes, as well as phytoplankton biomass change (ΔB multiplied by a factor of 10; solid line); bar plots of the integrated change over the decline phase due to (b) biological fluxes and (c) physical fluxes averaged over the focus region (PP: primary production; GRAZ: consumptive mortality; MORT: natural mortality; EXU: exudation; SINK: sinking; MIX: mixing; ADV: advection). A positive or negative sign of the bars here designates fluxes promoting and opposing an increase in MLD-integrated phytoplankton biomass, respectively. The magnitude (including the sign) of the integrated change is given as numbers above and below the bars.

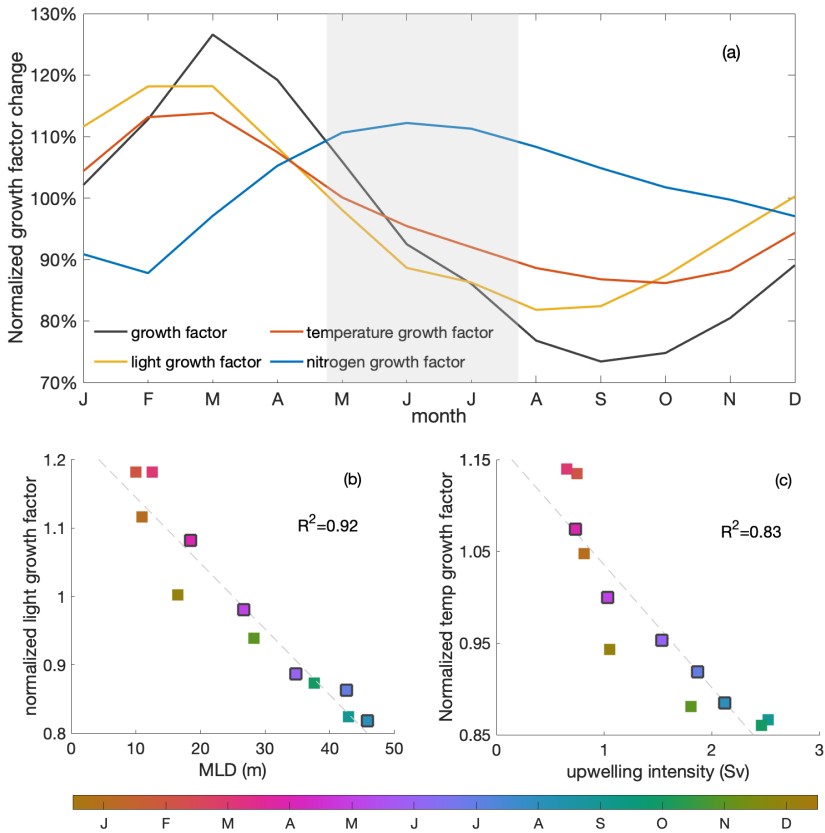

**Figure 6.** (a) Seasonal cycles of the normalised total (black) growth factor and light (yellow)-, temperature (red)- and nitrogen (blue)-related growth factors for phytoplankton over the mixed layer. The grey shading indicates the decline phase. (b) Seasonal correlation of MLD and mixed layer-averaged light-related growth factor; (c) correlation of upwelling intensity and mixed layer-averaged temperature-related growth factor. Colours indicate the time of the year (months), with black edges indicating the months of the decline phase. The $R^2$ values of the correlations are shown in the right-hand sides within the panels.

layers were deepest. The change of the temperature-related growth factor within the mixed layer was closely related with
the seasonal variation of upwelling intensity ($R^2 = 0.83$), with the lowest values occurring in September and October when upwelling intensity reached its maximum. During the decline phase, cold waters were upwelled into the mixed layer at a higher rate, further damping phytoplankton growth in addition to the effects of limiting light conditions. Reduced winter surface solar radiation and heat loss to the atmosphere also played a role in the seasonality of the light and temperature growth factors, respectively (Fig. C2), though to a much smaller extent (not shown) which agrees with findings in Echevin et al. (2008).

### 3.4.3   Enhanced upwelling and offshore transport of phytoplankton

An enhanced advective loss of mixed layer phytoplankton is a second-order process, promoting the decrease of MLD-integrated phytoplankton biomass during the decline phase (Fig. 5c). Similar to the effects of nutrients, phytoplankton biomass is affected by the seasonality of upwelling and offshore export of waters. Relatively dilute concentrations of phytoplankton growing below the base of the mixed layer are upwelled into the mixed layer, while waters with mixed-layer-averaged phytoplankton concentrations are pushed offshore. During the decline phase, the upwelling and offshore transport of water increased and a greater volume of what was produced in coastal waters was exported offshore: 4% of primary production was lost via advection by the end of the decline phase compared to 2% gained at the beginning. This greater loss of biomass due to divergent lateral advection was mainly caused by stronger upwelling during the decline phase (Fig. C3).

## 3.5   Seasonal Paradox: from phytoplankton to export

Small and large zooplankton exhibit the same "seasonal paradox" pattern as phytoplankton, and so does the export of organic material to the deeper ocean. Similar to phytoplankton, both small and large zooplankton are vertically well mixed within the mixed layer throughout the year (contours and colours in Fig. 7a, respectively). Biomass concentrations are high in austral summer and low in austral winter, in opposition to the upwelling trend. Additionally, the particulate organic matter, the sum of plankton biomass and other organic particles, follows the same pattern, with large amounts of particulate organic matter concentrated in a shallow mixed layer during the productive summer (Fig. 7b). The pattern of organic matter in the water column is then reflected in the export pattern of sinking organic material, composed of large phytoplankton as well as small and large detritus (Fig. 7c). Export below 100 m depth is high during the productive summer, when the mixed layer is shallow and particulate organic matter is large, and low in winter (Fig. 7b, black line). That is, the model's ecosystem is affected by the seasonal variation of the MLD.

Finally, export efficiency also follows the seasonal cycle of the MLD. Export efficiency is defined as the export of sinking organic material through the 100 m depth level, relative to primary production in the upper 100 m. It reaches a maximum in austral summer, when MLD is shallow, and a minimum in austral winter, when MLD is deep (Fig. 7d). As both export and primary production show the same seasonal trend as phytoplankton biomass, export must overcompensate the change in primary production and vary even more, in order to allow export efficiency to reveal the same seasonal trend. Export largely consists of large detritus originating from large zooplankton faecal pellets and mortality (Fig. 7c). Since large detritus possesses the fastest sinking speed compared to other components, it sinks the most efficiently. The relative contribution of fast-sinking large detritus to total export is largest in summer, close to 100 %, which may partially explain the higher export efficiency. In addition to changes in composition of the sinking organic material, other processes may cause export to be amplified relative to phytoplankton production. These include: (1) changes in structure and trophic transfer efficiency of the plankton food web, and (2) a varying degradation of sinking organic matter in the upper 100 m, that is, differences in the remineralisation. The detailed mechanisms behind the seasonality of export efficiency are beyond the focus of this paper and will be investigated in a separate study.

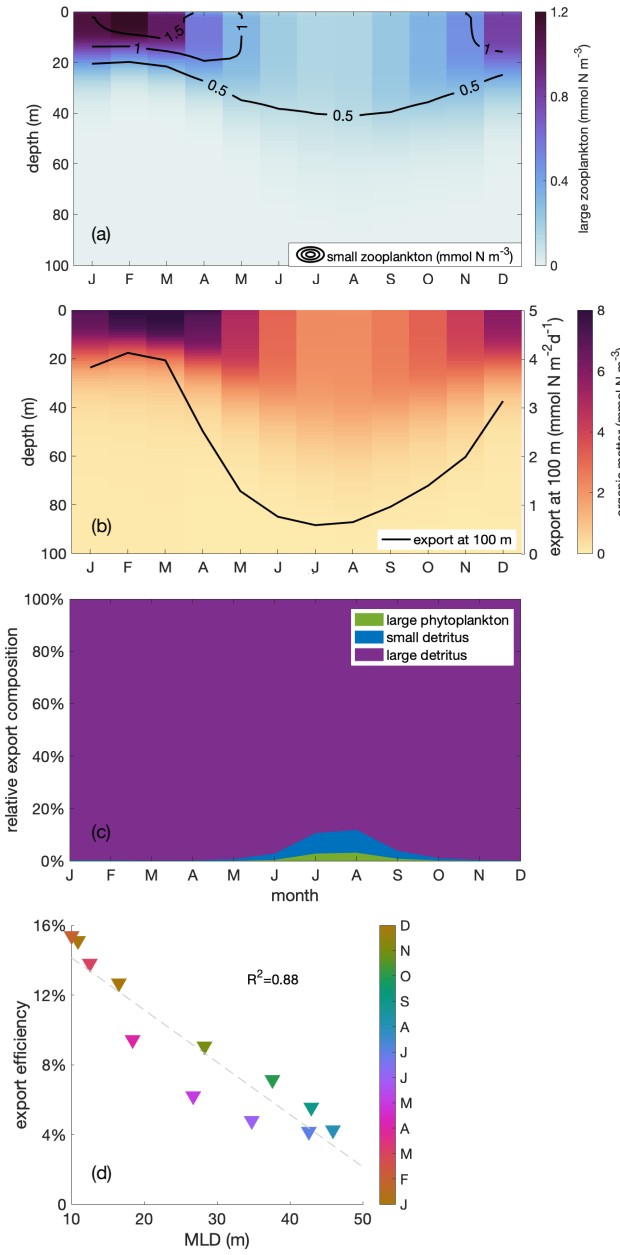

**Figure 7.** Monthly depth distribution of (a) large and small zooplankton (in colours and contour lines, respectively) and (b) organic matter with the seasonal cycle of vertical export through the 100 m depth horizon (black line); (c) Seasonal cycle of the relative contributions to sinking organic matter from large phytoplankton (green) and small (blue) and large (purple) detritus; (d) correlation of export efficiency with mixed layer depth (MLD) within the focus region. The export efficiency is defined as the ratio of export through the 100 m depth level to primary production in the upper 100 m. Colours indicate the month of the year.

## 4 Discussion

### 4.1 Mixed layer depth drives surface phytoplankton biomass seasonality in the Peruvian upwelling system

The regional ocean circulation-biogeochemical model that we used successfully reproduced the "seasonal paradox", defined as the seasonal out-of-phase surface chlorophyll concentration and upwelling intensity, as derived from observations. As shown in the results, the low surface chlorophyll concentration in strong upwelling conditions during austral winter was constrained by a combined effect of MLD-driven and upwelling-driven processes. Under strong upwelling conditions during austral winter, phytoplankton was diluted over a deeper mixed layer, leading to a decrease within the mixed layer. Likewise, surface 295 phytoplankton concentrations decreased by over 50%. Also, phytoplankton growth was slowed due to deteriorating light and temperature conditions, as well as strong upwelling pushing phytoplankton offshore.

Several previous studies have also focused on the possible reasons behind the seasonal paradox in the PUS. Echevin et al. (2008) also used a regional ocean circulation-biogeochemical model to examine the reasons for the relatively low surface chlorophyll concentration off the Peruvian coast in austral winter. Based on a series of model sensitivity experiments regarding 300 vertical mixing, surface temperature, iron limitation and insolation cycle, Echevin et al. (2008) concluded that the low surface chlorophyll in austral winter is mainly generated by the combined effect of dilution and deteriorating light with deepening mixed layers. Additionally, the iron sensitivity experiment confirmed the existence of iron limitation in austral winter, which corroborated the findings in Messié and Chavez (2015). Messié and Chavez (2015) pointed out that more severe iron limitation under low light could also be one of the reasons behind low primary production under strong upwelling conditions. According 305 to results from culture experiments, phytoplankton iron demand would increase under light limitation (Sunda and Huntsman, 1997). Based on observations, Friederich et al. (2008) suggested that winds in the strong-upwelling winter conditions favour curl-driven offshore upwelling, which would draw more offshore iron-deficient waters to the surface. On the contrary, a model study (Albert et al., 2010) found that stronger wind-curl-driven upwelling actually recruits more nutrient-rich water from a shoaling coastal undercurrent, thus enhancing surface chlorophyll concentrations. We could not assess the role of iron in 310 regulating the seasonality of phytoplankton biomass because our biogeochemical model did not simulate iron. Nevertheless, our study confirmed the importance of vertical redistribution of biomass and light limitation due to vertical mixing.

### 4.2 Upwelling into deep mixed layers: A unique feature of the Peruvian upwelling system and its implications

As stated in the previous paragraphs, based on the differences in the seasonalities of MLD and upwelling in the Peruvian system, upwelling of nutrient-rich waters occurs when growth conditions are least optimal, in particular when light availability 315 is lowest due to deep mixed layers. In contrast, within the California system, nutrients are upwelled into the shallowest mixed layers. While this nutrient supply coincides with shoaling mixed layers, associated with improved light conditions and reduced dilution, it does not result in as high a level of phytoplankton concentrations as for the Peruvian system. This supports that nutrient limitation contributes substantially to processes in the California system, as the supply of nutrients to shallow mixed layers through upwelling appears to be insufficient to relieve nutrient limitation. Additionally, if nutrients are upwelled into 320 deep mixed layers as in the Peruvian system and allow the onset of a bloom, zooplankton standing stocks might be low and

require more time to catch up, eventually reducing phytoplankton biomass. On the contrary, if nutrients are upwelled into shallow mixed layers, zooplankton standing stocks are likely already elevated, allowing zooplankton to immediately limit any increase in phytoplankton biomass.

While the Canary and Benguela systems lack a pronounced seasonality in MLD and phytoplankton, respectively, we point out a few aspects that may elucidate the role of MLD in these systems. Given that the Canary system does not feature a substantial seasonal MLD variability, it is intuitive that phytoplankton follows the seasonality of upwelling intensity more strongly compared to the other EBUS. While mixed layer conditions do not modulate the seasonality of phytoplankton, they may contribute to high phytoplankton concentrations in the Canary system insofar as mixed layers are shallow throughout the year, creating favourable light conditions. Finally, the Benguela system features a rather constant upwelling throughout the year into varying mixed layers. The unresponsiveness of phytoplankton to the varying MLD could hypothetically be due to compensating effects of deepening mixed layers that dilute phytoplankton and deteriorate light conditions, but are simultaneously accompanied by an enhanced supply of nutrients that are mixed up from below.

Other factors may also contribute to regulating phytoplankton in the EBUS aside from nutrients, dilution and light associated with upwelling and MLD, including the advection of biomass and regulation by temperature that varies with upwelling (also see the Results section; Messié and Chavez, 2015). Eddies have been found to favour offshore export and subduction of phytoplankton and nutrients (Lathuilière et al., 2010; Gruber et al., 2011; Messié and Chavez, 2015). In addition, Lachkar and Gruber (2011) suggest that a longer residence time because of a wide shelf and weak mesoscale activity may promote phytoplankton growth in the Canary system. Next to iron supply from the shelves and upwelling of source waters, Fung et al. (2000) also found that atmospheric deposition of iron varies between EBUS.

## 4.3 Seasonal paradox and ecosystem functioning

The interplay of mixed layer depth and upwelling that leads to the seasonal paradox in the PUS propagates further up the food chain, modulating the trophodynamics. In austral summer, the shallow mixed layer along with the add-on effect from upwelling supports the highest phytoplankton biomass and primary production, providing an ideal feeding place for zooplankton. In contrast, during the winter zooplankton face a food shortage, less efficient grazing due to dilution, and transport offshore due to enhanced upwelling. Similar to the spatial match-mismatch observed for phytoplankton and top predators in the Benguela system (Grémillet et al., 2008), mesozooplankton with its slower growth rate may also be negatively affected by enhanced upwelling.

In our model, mesozooplankton is responsible for the major part of the export in the coastal upwelling region. During the productive season, the faecal material of mesozooplankton accounts for close to 100% of the sinking matter, which is in good agreement with what Stukel et al. (2013) observed for the California system. We found that both primary production and export can be determined from the mixed layer dynamics and food web structures (consistent with Ducklow et al., 2001; Turner, 2015; Steinberg and Landry, 2017). The efficiency of the export, defined as the ratio of export to primary production, is also related to trophodynamics. We find that export efficiency is positively correlated with MLD on a seasonal scale. As mentioned in the "Results" section, it partially depends on the composition of the exported material. Mesozooplankton produce fast-sinking large

detritus, which enhances the export efficiency during the productive season. Kelly et al. (2018) observed that export efficiency is negatively correlated with net primary productivity in the California system. They suggested that the negative correlation in the California system arises from a seasonal decoupling of export and particle production through long-lived particles that introduce a temporal lag of mesozooplankton production and export to depth. Henson et al. (2019) also identify a negative correlation between export efficiency and primary productivity on a global scale. They imply in their study that not just the

phytoplankton community, but also the food web structure, is important to export efficiency. Currently, it is not entirely clear why the PUS export efficiency behaves differently in our model. We suggest that the interplay of the mixed layer and upwelling in EBUS and ecosystem functioning are closely linked, warranting further examination.

## 5 Conclusions and potential implications

In summary, CROCO-BioEBUS performs well with respect to observational data and successfully reproduces the "seasonal

paradox" with an out-of-phase relationship between surface chlorophyll and upwelling intensity in the Peruvian coastal waters. In agreement with an earlier model study (Echevin et al., 2008), the seasonal cycle of surface chlorophyll concentration in our simulations is driven mostly by MLD-related processes, specifically dilution and light limitation. Furthermore, our model results provide evidence for secondary contributions from upwelling-related processes such as temperature limitation and advection. This is consistent with Lachkar and Gruber (2011) and Messié and Chavez (2015), who suggested that advection

is relevant to the seasonal cycle, but in contrast to Echevin et al. (2008), who found that temperature was not important. Differences in results from Echevin et al. and our results likely originate in the different biogeochemical model components (e.g., different parameterisations of temperature dependencies of phytoplankton growth). Given the disparity of the models, the role of temperature limitation in the PUS warrants further investigations in order to better constrain second-order drivers of the seasonal paradox. The sensitivity of the different processes within the plankton ecosystem to temperature, as well as their

interplay, are topics of active research (e.g. Thomas et al., 2017; Chen and Laws, 2017; Morán et al., 2018; Marañón et al., 2018; Barton and Yvon-Durocher, 2019) and relevant particularly in light of global warming.

We find that the seasonal variability of phytoplankton propagates up the food chain and is reflected in trophodynamics and ecosystem functioning. In particular, zooplankton and organic matter within the water column mirror the seasonal cycle of phytoplankton. Finally, export and export efficiency are well-correlated with the MLD over the course of the annual cycle.

Given that changes in MLD are correlated to many ecosystem components related to plankton ecosystem functioning, we argue for a more thorough understanding of the impact of the seasonal paradox on the ecosystem. In particular effects on the trophic transfer of energy through the plankton food web to higher trophic levels such as fish will determine ecosystem functions like trophic transfer efficiency, fish production, and ultimately potentially fisheries yields. Thus, a better understanding of how the interplay of MLD and upwelling impacts the ecosystem in the contemporary PUS will ultimately help to better project how

coastal upwelling ecosystems, and in particular the Peruvian system, may vary under climate change.

Phytoplankton will inevitably be influenced by climate change, responding to changes in the biotic and abiotic environment. Impacts in a changing climate will arise from changes in stratification and upwelling, that further lead to shifting growth

conditions due to changes of light, temperature and nutrients (Behrenfeld, 2014). A recent regional modelling study (Echevin et al., 2020) projects a weak decrease in upwelling along with increasing stratification in the PUS due to climate change.

Our results suggest that the decreasing upwelling and increasing stratification will both contribute to an increase in surface phytoplankton, in agreement with the findings of Echevin et al. (2020). While a reduction of upwelling might lead to a reduced supply of nutrients, the region is far from being nutrient limited. Therefore, a reduction in upwelling could rather have an effect via temperature, reducing the cooling effect of upwelled waters. We hypothesise that the coastal region would experience more phytoplankton growth and biomass buildup with a reduction of upwelling, due to warmer surface waters and weaker offshore

advection compared to the current environmental situation. Moreover, according to our results, shoaling of the mixed layer will be more relevant than a decrease in upwelling intensity, reducing the dilution of phytoplankton and the light limitation in austral winter. This could possibly lead to an attenuation of the seasonal paradox in the future. As export and export efficiency are also regulated by MLD dynamics, we expect not only enhanced export but also an increase in the fraction of primary production that is transported to the deep ocean under global warming.

*Code availability.* CROCO and BioEBUS models are available at http://www.croco-ocean.org

*Data availability.* The model data used in this paper are available via the corresponding author

## Appendix A: Methods

### A1 Two-way nesting approach

Figure A1 visualises the coarser-resolution parent domain and nested finer-resolution child domain that contains the focus

region. The variables in section B are shown for the child domain.

### A2 Adjustment of biogeochemical model parameters

The parameter setting is the same as in José et al. (2017), with only a few biological parameters adjusted to make the ecology (phyto- and zooplankton biomasses, productivity) better fit observational data. The changed parameters along with value ranges from literature are listed in Table A1 and will be further explained below.

Here, we assign a higher mortality rate for large phytoplankton to simulate the potential impact of virus infection during bloom conditions (Suttle, 2005). Simulated phytoplankton biomass and its seasonality has been calibrated and evaluated against chlorophyll concentration data from MODIS monthly climatology data (https://oceancolor.gsfc.nasa.gov/). Nitrate has been evaluated based on World Ocean Atlas (WOA; Garcia et al., 2019) and cruise data (M92 and M93 Thomsen et al., 2016) while simulated MLD has been validated against the ARGO mixed layer database (http://mixedlayer.ucsd.edu/; Holte et al., 2017).

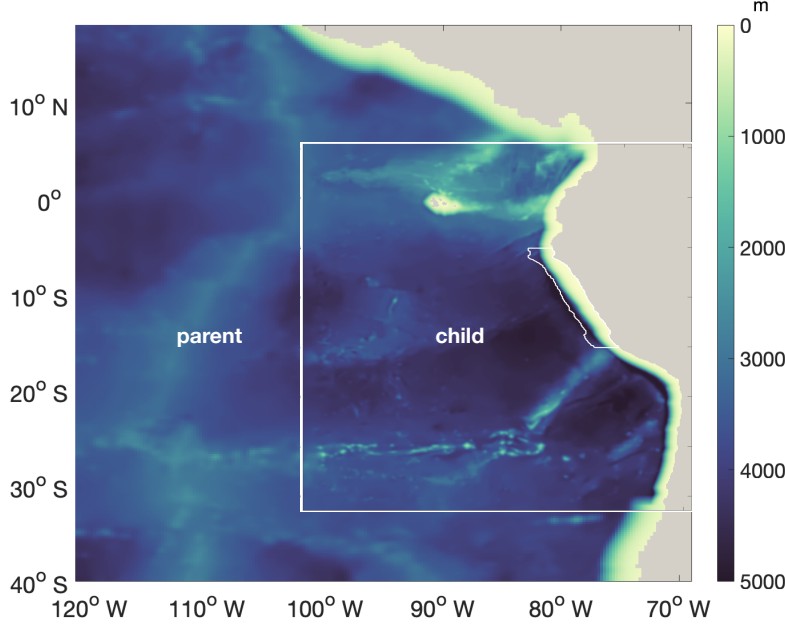

**Figure A1.** Bathymetry of the "parent" (1/4° resolution) and "child" (1/12° resolution) domains. White lines near the coast highlight the focus region.

## Appendix B: Model evaluation

### B1 Surface chlorophyll concentration

The large-scale spatial pattern of annual average surface chlorophyll of the monthly climatology of MODIS data and CROCO-BioEBUS are similar (Fig. B1), with higher chlorophyll concentrations in coastal regions and lower concentrations offshore
(note that chlorophyll is shown in log-scale). The satellite data features a higher cross-shore chlorophyll concentration gradient compared to the model simulation. The model's overestimation of the low offshore chlorophyll and hence weaker cross-shore gradient potentially is due to the lack of iron limitation in the model. Apart from that, the model is also not able to correctly capture the alongshore pattern (Fig. B1), i.e. it misses two observed high surface chlorophyll concentration patches between 8°S to 10°S and 12°S to 14°S (Bruland et al., 2005). Within a 200 km band near the coast, both satellite data and the model
simulation show a similar seasonality with maximum chlorophyll concentrations exceeding 4 mg /m$^3$ from March to April and minimum concentrations around 2 mg /m$^3$ in August. In general, simulated surface chlorophyll concentrations agree reasonably well with satellite data.

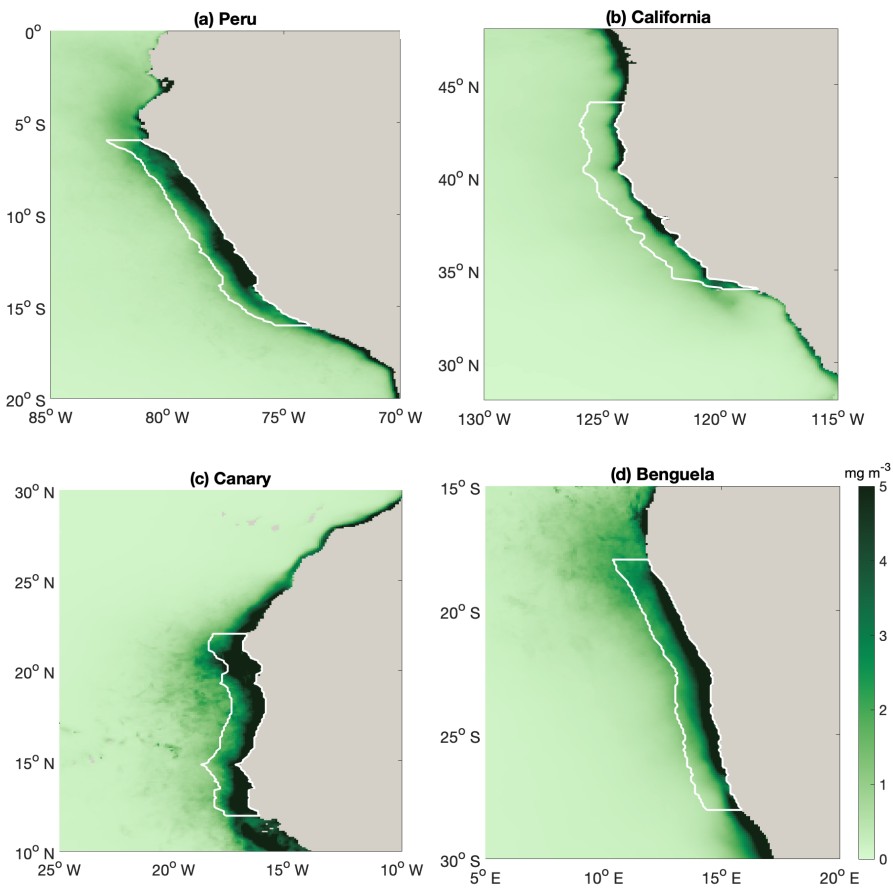

**Figure A2.** Map of annual mean surface chlorophyll ($mg\,chl\,m^{-3}$) with white lines highlight the regions that we average over in our analyses in Fig.2. Coastal EBUS regions picking here are the same as Chavez and Messié (2009)

## B2 Surface nitrate concentration

The simulated surface nitrate distribution shows the same seasonality as observations from the World Ocean Atlas (WOA; Garcia et al., 2019) (Fig. B2). The simulated surface nitrate concentration in the coastal region is biased high compared to the WOA data. This may be partly due to the WOA data failing to capture high-nitrate concentrations due to coastal upwelling. This notion is supported by nitrate concentration data from two cruises (Thomsen et al., 2016, M92 and M93) in austral summer that show nitrate concentrations in the coastal region are high compared to the model data.

**Table A1.** Adjusted biological parameters and range of published parameter values

| Parameters | Symbols | Units | Value | Range |
|---|---|---|---|---|
| Max growth rate of $P_L$ | $a_{P_L}$ | $d^{-1}$ | 0.6 | $0.6^a$-$3.0^b$ |
| Mortality rate of $P_L$ | $\mu_{P_L}$ | $d^{-1}$ | 0.15 | $0.027^c$-$0.2^d$ |
| Preference of $Z_S$ for $P_S$ | $e_{Z_S P_S}$ | - | 0.65 | see references[e] |
| Preference of $Z_S$ for $P_L$ | $e_{Z_S P_L}$ | - | 0.35 | see references[e] |
| Preference of $Z_L$ for $P_S$ | $e_{Z_L P_S}$ | - | 0.1 | see references[f,g] |
| Preference of $Z_L$ for $P_L$ | $e_{Z_L P_L}$ | - | 0.4 | see references[f,g] |
| Preference of $Z_L$ for $Z_S$ | $e_{Z_L Z_S}$ | - | 0.5 | see references[f,g] |
| Excretion rate of $Z_S$ | $\gamma_{Z_S}$ | $d^{-1}$ | 0.1 | $0.03^h$-$0.1^i$ |
| Excretion rate of $Z_L$ | $\gamma_{Z_L}$ | $d^{-1}$ | 0.1 | $0.05^h$-$0.1^i$ |
| Mortality rate of $Z_L$ | $\mu_{Z_L}$ | $\mathrm{mmol\,N\,m^{-3}\,d^{-1}}$ | 0.135 | $0.05^a$-$0.25^j$ |

The values for diet preferences were picked based on a combination of calibrating the model against observations of plankton biomasses and observed qualitative diet preferences in the references.

[a] Gutknecht et al. (2013)

[b] Andersen et al. (1987)

[c] Koné et al. (2005)

[d] Taylor et al. (1991)

[e] Bohata (2016)

[f] Kleppel (1993)

[g] Schukat et al. (2014)

[h] Aumont (2005)

[i] Fennel et al. (2006)

[j] Lima and Doney (2004)

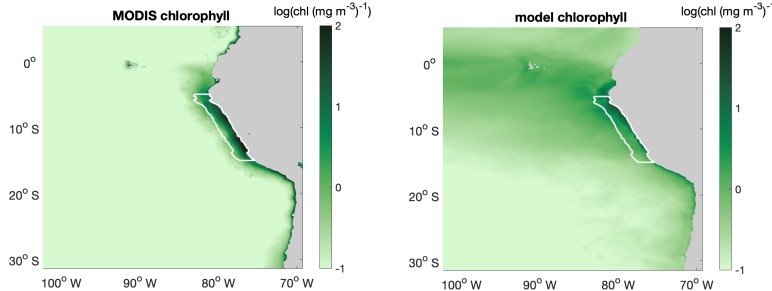

**Figure B1.** Annual mean surface chlorophyll concentration (in $\log(\mathrm{chl}\,(\mathrm{mg\,m^{-3}})^{-1})$) distribution of (a) MODIS and (b) CROCO-BioEBUS. White lines highlight the focus region.

## 435 B3 Mixed layer depth

We validate the simulated MLD against both the gridded ARGO mixed layer dataset (Holte et al. (2017), http://mixedlayer.ucsd.edu/) and the de Boyer Montégut climatology mixed layer data available from the IFREMER/LOS Mixed Layer Depth Climatology

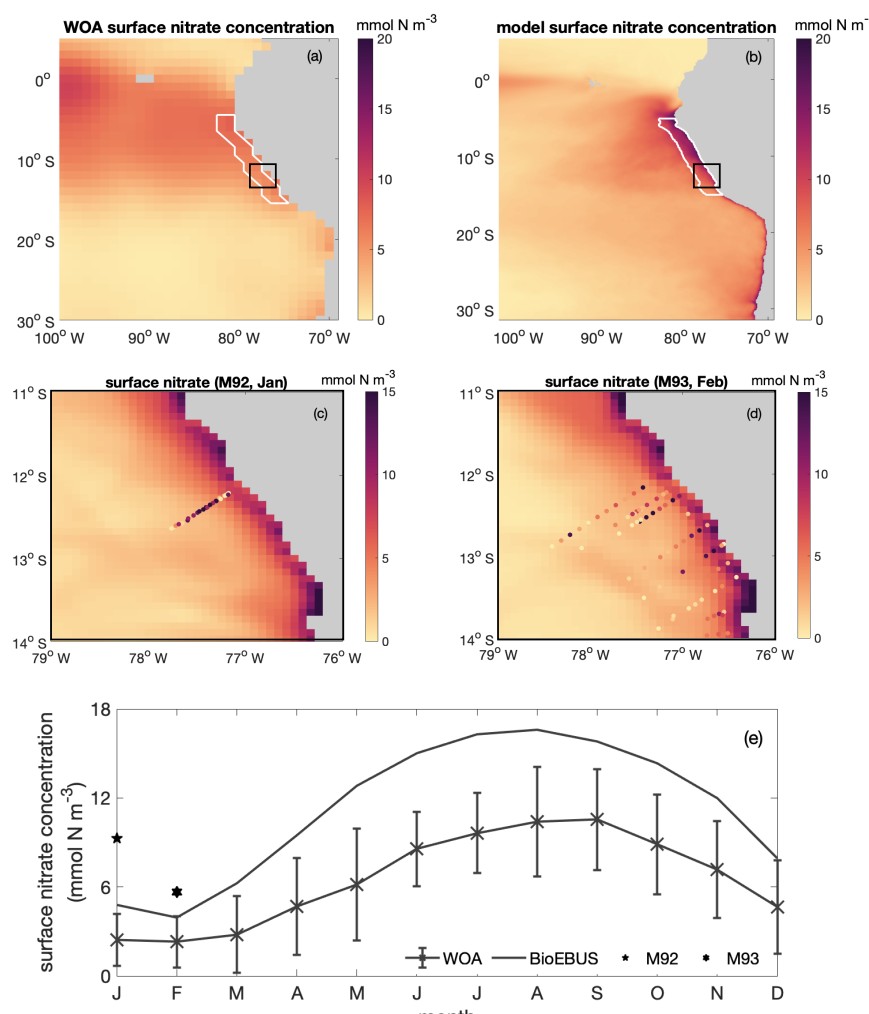

**Figure B2.** Spatial distribution of surface nitrate concentration based on (a) WOA and (b) CROCO-BioEBUS; (c) January and (d) February as simulated by CROCO-BioEBUS. Dots indicate measurements from the cruises M92 (January) and M93 (February); (e) Seasonal cycle of surface nitrate concentration from WOA (cross), CROCO-BioEBUS (line) and cruises (pentagram, hexagram) within the focus region. White lines highlight the focus region. The black box indicates the maps of panel c-d.

website (www.ifremer.fr/cerweb/deboyer/mld) within the research area (Fig. B3). All observational data and simulated MLD are calculated as the depth where a $0.2^o C$ difference to the surface temperature is reached. The annually averaged spatial dis-
tribution of MLD within the research area presents the same features as ARGO: shallower MLD in the coastal region (around 20 m) and deeper MLD in the offshore region (around 80 m). The simulated seasonal variability of MLD within the research region generally follows the seasonal trend of the ARGO and the Boyer Montégut climatology data. The water column within

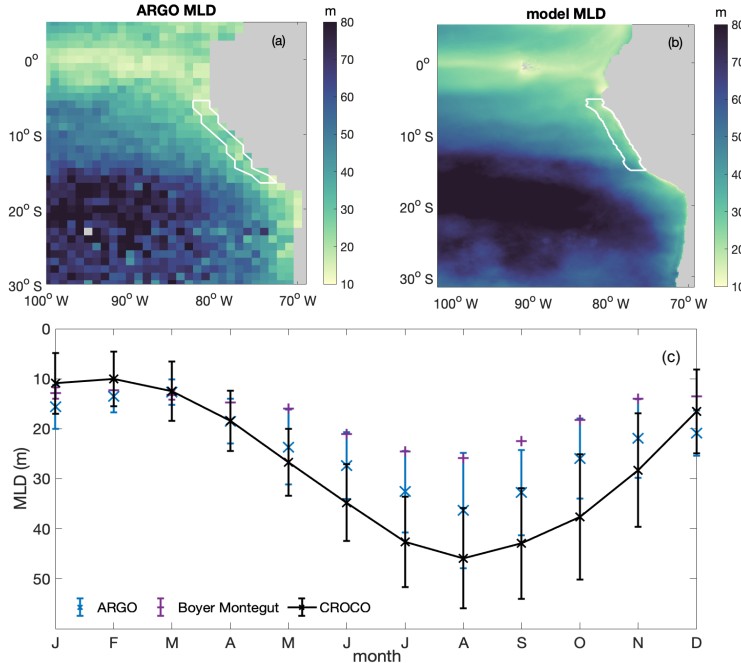

**Figure B3.** Annual average spatial distribution of mixed layer depth (MLD) from (a) ARGO and (b) CROCO-BioEBUS; (c) Seasonal variation of average mixed layer depth from ARGO (blue), the de Boyer Montégut climatology (purple) and model simulation (black) with error bar indicating the standard deviation within focus region. White lines highlight the focus region.

the research region is most stratified in February to March and most deeply mixed in August. Although simulated MLD in austral winter is somewhat deeper than Argo and the de Boyer Montégut climatology data, the simulated MLD and the de Boyer

Montégut climatology data are largely within the range of the ARGO data. Deeper simulated MLD compare to observation could partially come from not including chlorophyll shading effect on water cooling (Echevin et al., 2021).

## B4 Sea surface temperature

The simulated SST has been validated against monthly climatological MODIS data in terms of both spatial pattern and seasonal variability within the research area (Fig. B4). The annually averaged spatial distribution of SST is well simulated by the model.

The model successfully captures the cold coastal upwelled water as well as slightly warmer water masses further offshore. The simulated SST seasonality within the research region generally follows the seasonal trend of the observations, with a cool bias of less than $1^{o}C$. The surface waters within the research region are warmest in February to March matching the modelled/observed shallowest mixed layers and coldest from August to October. In general, the simulated SST matches the observations well both in terms of spatial pattern and seasonal variation.

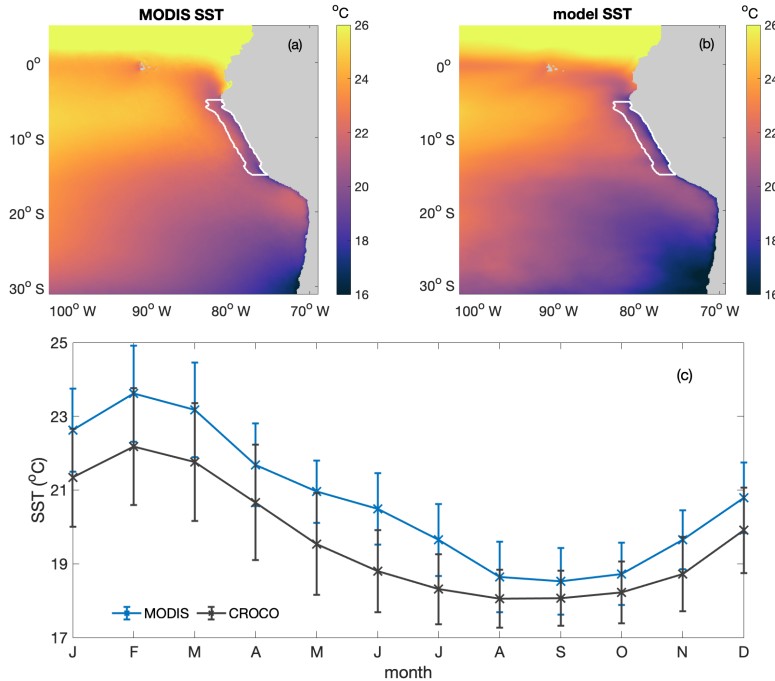

**Figure B4.** Annual average spatial distribution of sea surface temperature (SST, in $^{o}C$) from (a) MODIS and (b) CROCO-BioEBUS; (c) Seasonal variation of average sea surface temperature (SST) from MODIS (cross) and model simulation (line) within focus region. White lines highlight the focus region.

## B5  Mesozooplankton distribution

In addition, we calibrated zooplankton in the BioEBUS model against observational estimates (Fig. B5). Calibration and assessment of simulated zooplankton is often omitted, despite the central role of zooplankton parameterisations on plankton dynamics (Anderson et al., 2010; Prowe et al., 2012). While the observations show a large spread, the simulated large-scale spatial distribution of mesozooplankton generally follows the observed pattern, with high mesozooplankton biomass in the upwelling region and low biomass further offshore. The overestimated simulated zooplankton biomass compared to the observational data in the offshore region is likely partially related to the overestimated offshore phytoplankton biomass, which in turn presumably results from the lack of iron limitation in the model. As shown in Fig. B5c, most data points fall close to the 1:1 line. However, the model is not able to capture the few data points with very high zooplankton biomass. The model simulates a stripe of low zooplankton biomass concentrations in the focus region near the coast (due to offshore advection combined with slow mesozooplankton growth) that is difficult to assess, as observations near the coast are sparse. This feature may be apparent to some extent in the observations in the southern focus region. Note that observational zooplankton biomass estimates are based on a wide range of methods and accordingly have a large uncertainty that is difficult to quantify (O'Brien, 2007). An agreement of model and observations in magnitude and large scale pattern is therefore a meaningful result.

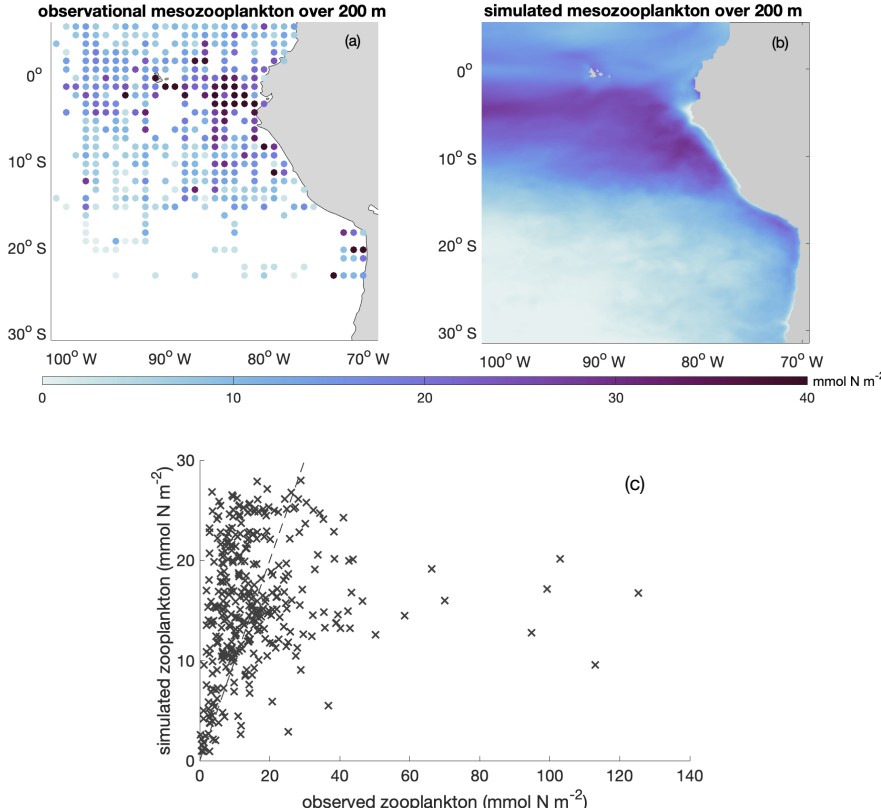

**Figure B5.** Annual average spatial distribution of integrated mesozooplankton biomass over upper 200 m based on (a) observational data (Moriarty and O'Brien, 2013, https://doi.org/10.5194/essd-5-45-2013) and (b) model simulation. (c) Scatter plot of observed and simulated integrated mesozooplankton biomass over upper 200 m (in $mmol\ N\ m^{-2}$). The dashed line indicates the 1:1 line.

## Appendix C: Additional Figures

The whole time series of temperature and nitrate concentration at 10 m and 100 m are shown in Fig. C1a-b. Surface fields are spun up after one year while water at 100 m takes 3-10 years longer to reach a steady state. In the meanwhile, mixed layer and surface layer chlorophyll are also spun up after one year (Fig. C1c-d).

    Apart from above mentioned mixed layer depth and upwelling intensity, short-wave surface radiation and surface net heat flux are of second-order importance to light- and temperature-related variance during the decline phase respectively (Fig.C2).

Phytoplankton net advection flux over the mixed layer closely follows the upwelling intensity during the decline phase (Fig.C3, $R^2 = 0.81$). When the mixed layer depth is relatively shallow, the correlation between upwelling intensity and phytoplankton convergence of advection over the mixed layer is insignificant.

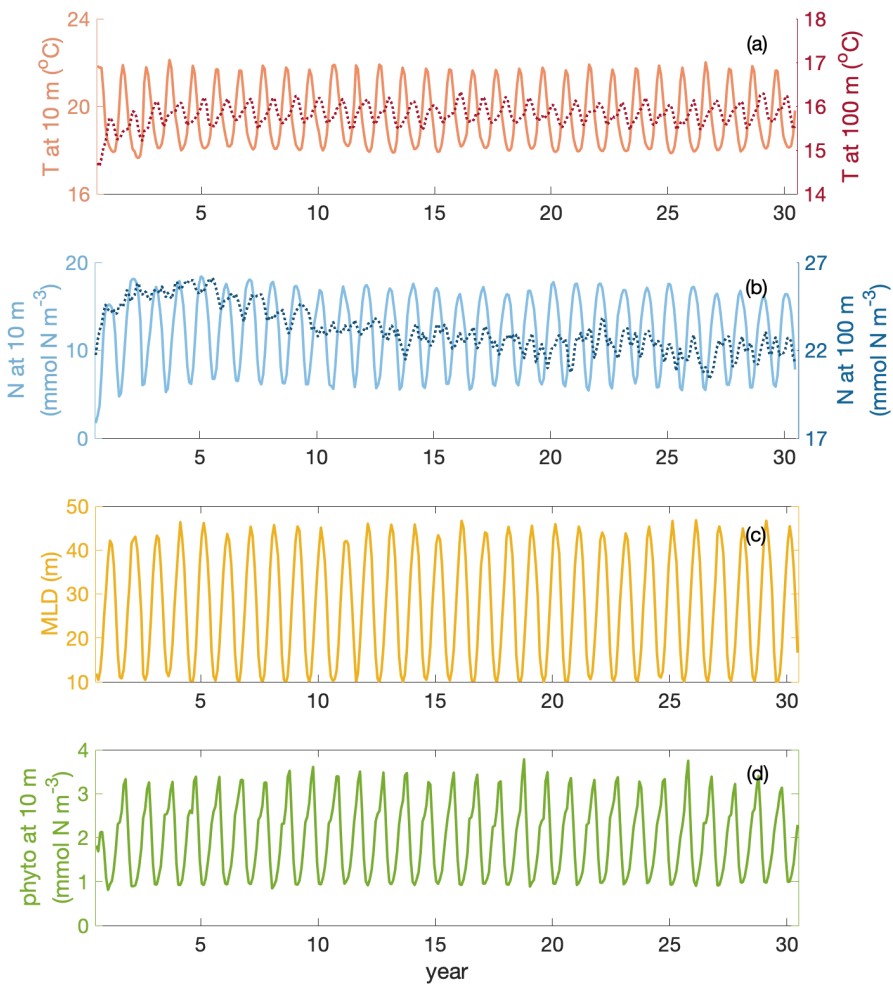

**Figure C1.** Time series of temperature T (at 10 m & 100 m depth), nitrate N (at 10 m & 100 m depth), mixed layer depth MLD and phytoplankton phyto (at 10 m) over 30 years of simulation in the focus region.

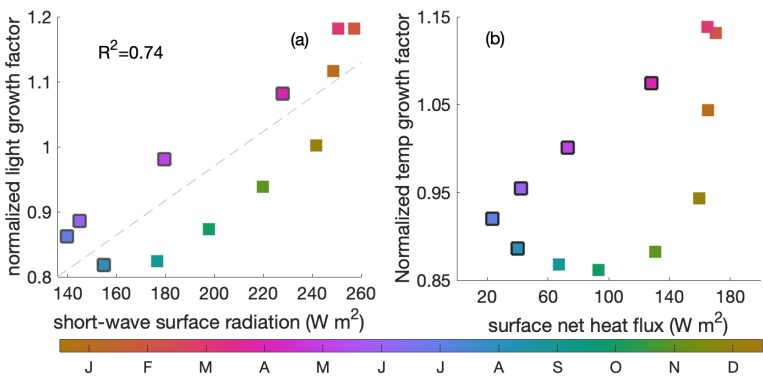

**Figure C2.** (a) Correlation between surface short-wave radiation ($Wm^{-2}$) and the averaged light-related growth factor within mixed layer; (b) Correlation between the surface heat forcing (in $^\circ Cd^{-1}$) and averaged temperature-related growth factor within mixed layer. Colour indicates the time of the year and black edges the decline phase.

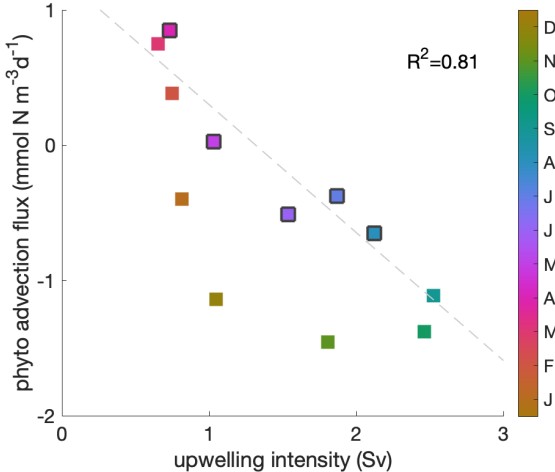

**Figure C3.** Correlation between upwelling intensity and phytoplankton convergence of advection over the mixed layer. A negative convergence equals a divergence of phytoplankton biomass due to the combined effect of upwelling and lateral transports. Color indicates the time of the year and black edges the decline phase. The correlation coefficient ($R^2$=0.81) is shown for the decline phase.

*Author contributions.* IF and AO designed the study. TX and YSJ carried out the simulations. TX, IF and FP conducted the analysis. All authors discussed the results and wrote the manuscript.

*Competing interests.* The authors declare that they have no conflict of interest.

*Acknowledgements.* This work is financially supported by the China Scholarship Council (TX, grant no.201808460055). Further support for this work was provided by the BMBF funded projects Coastal Upwelling System in a Changing Ocean CUSCO (IF, AO) and Humboldt Tipping (YJ).

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
