# Peer review of "Mixed layer depth dominates over upwelling in regulating the seasonality of ecosystem functioning in the Peruvian Upwelling System"

_Biogeosciences, 2021_

## Author Comment (AC1)

[Figure]

Figure 1: Organic matter ($mmol\,N\,m^{-3}$) depth-month distribution with the seasonal cycle of export efficiency at 100 $m$ (black triangle) within the focus region.

---

## Author Comment (AC2)

[Figure]

Figure R2.1: Comparison of the on- and offline calculated tendency of ML integrated phytoplankton biomass (mmol N m$^{-2}$ d$^{-1}$)

[Figure]

Figure R2.2: Maps of annual mean surface chlorophyll ($mg\,chl\,m^{-3}$) that show the EBUS regions that we focus on; white lines highlight the regions that we average over in our analyses.

[Figure]

Figure R2.3: Seasonal cycles of (a) total phytoplankton biomass change ($\Delta$B; in mmol N m$^{-2}$ d$^{-1}$); gray shading indicates the decline phase, with t1 and t2 marking the beginning and end of the decline phase; (b) phytoplankton fluxes combined to net biological gain (bio) and net physical loss (phy, in mmol N m$^{-2}$ d$^{-1}$, as shown in Eq. 1 plus the entrainment and detrainment introduced by the temporal variation of the MLD); (c) Stacked bar plots of balancing budgets at t1 and t2. pp stands for primary production; graz for consumptive mortality; mort for natural mortality; exu for exudation; sink for sinking; mix for mixing; adv for advection and entr for entrainment. Fluxes are distinguished as sources (positive values) and sinks (negative values) of phytoplankton biomass, with the sum of all source and sink terms balancing at times t1 and t2. All fluxes are integrated over the MLD.

[Figure]

Figure R2.4: (a) Seasonal cycle of phytoplankton source and sink processes (in color, stacked), as well as phytoplankton biomass change (ΔB, multiplied by a factor of 10; solid line). Bar plots of the integrated change over the decline phase due to (b) biological fluxes and (c) physical fluxes averaged over the focus region. Red shading indicates fluxes that promote an increase of ML-integrated phytoplankton biomass, and gray shading fluxes that oppose an increase. The magnitude of the integrated change is given as numbers on top/below the bars. pp stands for primary production; graz for consumptive mortality; mort for natural mortality; exu for exudation; sink for sinking; mix for mixing; adv for advection and entr for entrainment. All fluxes are integrated over the MLD.

$$\frac{\partial C}{\partial t} = BIO(C) + PHY(C) \qquad \text{(R2.1)}$$

with $[BIO = PP - GRAZ - MORT - EXU - SINK; PHY = MIX + ADV]$

---

## Author Comment (AC3)

[Figure]

Figure R1.1: Time series of temperature T (at 10 m & 100 m depths in solid and dashed, respectively), nitrate N (at 10 m & 100 m depths, in solid and dashed, respectively), mixed layer depth MLD and phytoplankton phyto (at 10 m) over the simulation period of 30 years in the focus region.

[Figure]

Figure R1.2: Top: Number of Argo profiles per 1x1 degree grid point. The white line highlights the focus region; Bottom: Seasonal variation of the mixed layer depth averaged over the focus region from Argo (blue), de Boyer Montégut climatology data (yellow) and the model simulation (red). The bottom and top edges of the box indicate the 25th and 75th percentiles, respectively. The whiskers extend to the most extreme data points not considered as outliers, and the outliers are plotted individually using the '+' symbol.

[Figure]

Figure R1.3: Seasonal cycles of surface chlorophyll concentration from the model simulation (black solid line, with bars indicating the spatial standard deviation, and Chl:N=0.795 to convert the model nitrogen output to chlorophyll), satellite data (dotted lines; SeaWIFS (diamond) and MODIS (square), bars indicate the spatial standard deviation) and in situ data (digitized from Pennington et al. (2006, star) and Echevin et al. (2008, cross), with bars indicating the estimates of uncertainty of the in-situ data).

[Figure]

Figure R1.4: Seasonal cycles of surface light (yellow) and temperature (red) growth factor.

[Figure]

Figure R1.5: Map of annual mean surface chlorophyll ($mg\,chl\,m^{-3}$) with white lines highlight the regions that we average over in our analyses.

[Figure]

Figure R1.6: (a) Seasonal cycles of phytoplankton (green), small zooplankton (yellow), large zooplankton (purple) biomass integrated over the upper 100 m and mixed layer depth (MLD, grey). Note that the vertical axes are scaled to range the annual mean plus/minus 100% for each of the variables. (b) Seasonal cycle of the sinking fluxes (sink) of large and small detritus and large phytoplankton, and (c) correlation of export efficiency with MLD. The export efficiency is defined as the ratio of primary production in the upper 100 m over export (sum of the sinking fluxes shown in panel (b)) through the 100 m depth level. Colors indicate the month of the year.

$$PP = C \cdot J_{max} \cdot L_{(PAR)} \cdot L_{(T)} \cdot L_{(N)} \qquad \text{(R1.1)}$$

$$J_{mld} = J_{max} * L_{mld} \qquad \text{(R1.2)}$$

$$\frac{dz}{dx} = \frac{dy}{dx} \cdot \frac{dz}{dy} \qquad \text{(R1.3)}$$

---

## Author Response (AR1)

Reply to reviewer #1

We would like to thank you again for your constructive comments. According to your comments, we elaborated on existing studies and expanded on the new aspects of our study, in particular with regard to zooplankton and ecosystem functioning. Your comments have helped us set apart our manuscript more clearly from previous studies. A detailed point-by-point response to your detailed comments is listed below:

**_Abstract:_**

L7: *"Intense upwelling coincides with deep mixed layers": is this really unique? Are the layers deep?*

R: We find that the pattern of intense upwelling coinciding with deep mixed layers in the seasonal cycle is unique to the HUS. As shown in Fig. 2e of the manuscript, the Humboldt system is the only of the four major EBUS that reveals a positive temporal correlation between MLD and upwelling intensity. We reworded the sentence (L6-7).

L11: *"In contrast to previous studies, reduced phytoplankton growth due to enhanced upwelling of cold waters and lateral advection are second-order drivers of low surface chlorophyll concentrations": not sure what previous studies were asserting.*

R: We have rephrased this section to be more precise about previous findings regarding lateral advection and temperature limitation (L12-15).

L15: *what about the role of nutrient enrichment? It could be reduced under climate change. See Echevin et al. (2020).*

R: We agree that nutrients could be reduced under climate change due to increasing stratification and potentially weakening upwelling. As discussed in our manuscript (L154-155 and L384-385), our results suggest that the system in its present state is not nutrient-limited. Although surface DIN concentration shows a clear seasonal cycle, it is not the limiting factor of phytoplankton growth, even when concentrations are lowest in summer. Unless future nutrient concentrations drop substantially below 'summer levels', we expect that other limiting processes (dilution, light condition) would dominate over the impact of reduced nutrients. Therefore, a dominant role of reduced nutrient supply in case of a weakening of upwelling seems less likely than other effects of increasing stratification.

**_Introduction:_**

L33-35: *The seasonal paradox is also mentioned by EC08. At citing the latter, it would be fair to mention that they used a regional coupled physical-bgc model as in the present study. You should also mention Thomas et al. (2001) and Chavez et al (2005) who noted the seasonal paradox.*

R: We have provided a more detailed description of EC08 and also included Thomas et al. [2001] and Chavez [1995] in the introduction (L34-45).

L38: *Actually EC08 showed that the seasonal cycle of insolation did not play a major role in their model set-up, whereas Guillen and Calienes (1981) assumed it played a role.*

R: We rephrased and explained this part more clearly (L42-44).

L39: *I disagree here: this has been at least partly assessed in EC08. Rephrase and explain precisely which hypotheses have not been assessed in the latter and which new ones will be assessed in the present work. Overall, a more accurate review of the findings of EC08 to explain the out-of-phase seasonal cycle of chlorophyll is needed in the introduction.*

R: As mentioned above, we have recapped the study of EC08 in more detail and emphasised more clearly the differences between our study and EC08 (L36-47).

L41: *"the unique mechanism": not clear what this unique mechanism is.*

R: The unique mechanism refers to the first question ("what is the uniqueness of PUS compared to other EBUS that leads to the seasonal paradox?"). And here it represents "the deep MLD coinciding with strong upwelling intensity". We have modified the sentence (L49-50).

**Data and methods**

L50: *why do you cite ROMS here? Explain that CROCO is the next generation ROMS-AGRIF model.*

R: The citation here is according to what is suggested on the CROCO website (see https://www.croco-ocean.org/how-to-cite/). We have added "CROCO is the next generation of the ROMS AGRIF model" and a reference in the manuscript (L57).

L54: *do you use bulk forcing*

R: No, we did not use bulk forcing but a forcing file (croco_frc) created with the croco-tools.

L61: *I think the authors should explain why these variables are important for the simulation of chlorophyll. Fig. 1 of José et al shows that the PCC is too strong in the model at 12°S, not too weak!*

R: As detailed features of the circulation are not the focus of this study and to keep the paragraph concise, we have rephrased the sentence to a general statement, referring to José et al, and saying that the model is able to reproduce the current system of the region reasonably well (L68-69).

L66: *I do not think it can be said that N is a species. I think the authors could have done a better job at proofreading the English in the submitted paper.*

R: The expression "N species" refers to different forms of nitrogen in the water and is commonly used by chemists. Take, for example, the title of Michalski et al. (2006): 'Determination of Nitrogen Species (Nitrate, Nitrite and Ammonia Ions) in Environmental Samples by Ion Chromatography'. To simplify the reference to nitrate, nitrite and ammonium we stick to summarise them as N species.

L72: *Table A1 is useful. Please indicate values in the table instead of "see ref."*

R: The parameters for which values are given as 'see refs' concern zooplankton diet preferences. The references we cite are biological studies based on observations, and the parameter values in the model orient themselves on these observations of stomach contents of individual taxa. As model parameters need to integrate over different taxa and their traits and feeding strategies, their values do not directly correspond to observed values of individual taxa. In our simulation, diet preference values are estimated based on the observed preference in the references and model calibration. Therefore, we prefer not to list specific values but a footnote, to avoid misleading the readers .

L73: *Good to know that the model has been adjusted to fit zooplankton biomass, which to my knowledge, has not been done before: which figure and section is it?*

R: We included the model evaluation against observed zooplankton biomass in the appendix (Fig. B5), along with an additional section with results regarding ecosystem functioning (section 3.5).

Figure C1: *I think showing N at 100 m is more relevant than at 1000m, as it would be in the depth range of upwelled waters. Besides, N at 1000m is not at quasi equilibrium after 25 years, it is still drifting quite a lot.*

R: Thank you for the advice. We replaced the N at 1000 m with N at 100 m (Fig. C1). N at 100 m is equilibrated well before the analyses period of the last five years.

L82: *"analyze with...": another typo. There are too many of them, the language really needs to be corrected, I am sure the authors can do a better job at it or get some help from a native speaker.*

R: We have switched to the British spelling variations in the revised manuscript.

L90: *add an equation for J and the limitation terms L(x).*

R: We have updated Equ. 2 in the revised manuscript.

L95: *I do not understand equation (3). What is $L_{mld}$ and how is it related to J? Does it play a role in the model equations? Please clarify.*

R: $L_{mld}$ represents the average growth rate phytoplankton has within the MLD. We have updated Equ. 2 and included the explanation in the revised text (L107-109).

L96: *Cmld is the concentration or the concentration change? If it is the concentration change, what is $\Delta$Cmld? I am lost here.*

R: Cmld refers to the average phytoplankton biomass concentration in the mixed layer. $\Delta$Cmld refers to the change of the concentration over time. We have corrected the symbols in the revised manuscript (L111-113).

L98: *What is the chain rule?*

R: We refer here to the mathematical expression "chain rule", the formula to compute the derivative of a composite function (Equ. R1.3).

Figure B3: *This is an interesting figure: could you indicate the number of data per 1x1 grid point? How does it compare to the Boyer Montegut climatology (can be downloaded here: http://www-.ifremer.fr/cerweb/deboyer/mld/home.php). It would be nice to add an error bar indicating the model's MLD internal variability.*

R: Thank you for pointing us to the additional data set. Please see Fig. R1.2 for the number of Argo float profiles. Fig. B3c now shows comparison of different data sets and the simulated MLD with error bar.

Figure 1d: *I am surprised the model chlorophyll is low: is this really the best fit after parameter tuning? Could you add an "error" bar to represent the model internal variability of Chl? Also, I think that Pennington's plot was not exactly the same coastal area as shown in Fig.1a.*

R: One reason for the simulated Chl being relatively low is that we used the Chl:N ratio from the model's croco-tool (the Matlab toolbox as mentioned above) to convert from nitrogen to chlorophyll (Chl:N=0.795). If we use the Chl:N ratio of Gutknecht et al. [2013] (Chl:N=1.59), the simulated Chl would double its value. The Chl:C and C:N ratios are both very variable in nature and it is not possible to resolve this variability with a fixed-stoichiometry-model as used here. Therefore, we tuned the model chlorophyll so that the observed chlorophyll values fall between the simulated chlorophyll values using the different Chl:N ratios. We have added the error bars in Fig. 1d. And thank you for mentioning that in Pennington et al. [2006] the data is taken for a 250 km band off the coast which now has been clarified in the revised manuscript.

**Results**

L137: *"suggested" is a bit weak here. See my previous comment about the introduction.*

R: We have changed it to "as found by Echevin et al. (2008)" in the revised manuscript.

L138: *"the weaker increase...": you need a reference here.*

R: The reference is the Peruvian system. This part has been rephrased in the revised manuscript.

L141: *Are the ARGO MLD estimates reliable off Peru? This should be addressed somewhere and ARGO data should be described in the Data and Methods section.*

R: As now shown in Fig. R1.2 (see response above), there are on average 120 profiles available summed up over the focus region every month, which allows to robustly estimate the seasonality of the MLD. We have added information on the Argo data and on all other data that are used in the results in the Data and Methods section (L121-126).

L142: *I think there are other hypothesis supporting the high chlorophyll values in the Canary system. This section comparing the different systems is too long and does not focus enough on the main topic of the paper.*

R: We have shortened and streamlined this section and focus more on the main topic of the paper.

L145-146: *I do not agree with the conclusion here: as SST is also strongly correlated with insolation, you can not conclude that the seasonal cycle of temperature drives the phytoplankton growth.*

*For your information, EC08 showed that the temperature effect was negligible in their model set-up (see end of section 3.3).*

R: We agree that a correlation does not imply causality, that is the relationship of SST and Chl does not necessarily mean that a higher SST is causing Chl to increase, because SST and Chl could commonly be forced by another variable, namely insolation. However, based on (i) previous findings by EC08 and (ii) our model analyses, we ruled out the contribution of insolation.

(i) EC08 showed based on a sensitivity study that in their model the light limitation due to the seasonality of insolation was weak (see their section 3.5).
(ii) In Fig. R1.4, the surface light and temperature growth factors show different seasonal patterns. In addition, the temperature growth factor reveals a greater relative seasonal variation than the surface light growth factor which would more directly reflect insolation.

*Figure 2: the regions where data is averaged in a coastal band in the four systems should be indicated in the supplementary.*

R: We have added Fig. A2 in the appendix to indicate the EBUS regions that were used to produce Fig. 2 in the manuscript.

*L150: The reason for this is well known: the along-shore wind forcing is enhanced during winter, increasing upwelling and vertical mixing, and the lower winter insolation decreases surface stratification and increases the MLD.*

R: We agree that the reasons for stronger upwelling and deep MLD in winter are well known. We here describe the relationships that we find in Fig. 2e and emphasise that the Peruvian EBUS is the only EBUS that shows such a positive correlation between MLD and upwelling.

*L181: L181: "DV contributes...": in which figure is this shown?*

R: The contribution of $\Delta V$ is calculated based on Equ. 5, and the effect of the dilution that goes along with the volume change is evident from Fig. 3b-c.

*L206: "advection is picking up": I can not see that, advection seems negligible with respect to mixing (Fig.4d).*

R: We noticed that Figure 4 contained a lot of information and was not discussed in sufficient detail in the manuscript to be easily digestible, as also indicated by the other reviewer. We have split up the original Fig. 4 into two figures (Fig. 4-5) and have added additional explanation to make the figures easier to understand.

*L208-209: "the decreasing rate...": I do not understand this sentence*

R: Section (3.4.1) regarding phytoplankton budgets and the discussion of Fig. 4-5 have been rewritten.

*L202-2011: This paragraph is very difficult to follow and lacks precise references to the figures in the core of the text.*

R: See above, we have split up the original Fig. 4 into two figures (Fig. 4-5) and have restructured the paragraph to make it easier to follow.

L222: *How do you obtain this 60% decrease based on Eq.3?*

R: We used the start of the decline phase as the reference, and checked how the three growth factors changed until the end of the decline phase. Then we used the mathematical product rule to calculate how each growth factor was contributing to the total change of the growth factor.

L229: *The weak role of temperature is in agreement with EC08.*

R: While in EC08 the role of temperature was concluded to be negligible, in our model it is, while minor, not negligible. That is, we concluded that the sensitivity of phytoplankton (and the ecosystem) to temperature and potential temperature changes could be model dependent and might be worthwhile to be investigated further.

**Discussion**

L258-261: *I suggest a closer examination of EC08 findings and expand the comparison with their modelling work, which is very similar to what is presented here. In particular, they relaxed iron limitation in their model and found an Chl increase of 20-80% (depending on the latitude) during winter and spring, which corroborates the impact of iron limitation on the seasonal cycle found by Messié and Chavez (2015).*

R: We have expanded the findings by EC08 and included a discussion of the results of the Fe-sensitivity study (L293-298).

L272: *the sentence seems incomplete.*

R: Corrected (L309-311).

L274: *"and in deep ...": rephrase*

R: This part has been deleted after streamlining.

L275: *"charge"*

R: This part has been deleted after streamlining.

L276: *I am not convinced by this hypothesis: the residence time of the upwelled water in the mixed layer near the coast is probably quite short as upwelled waters are rapidly transported offshore by Ekman currents. Thus I do not believe in such preconditioning. Unless you can you prove it using the model.*

R: We have removed this section.

L295: *"higher" with respect to what? Clarify.*

R: The sentence has been rephrased in the revised manuscript (L324).

L314: *is this a result of the study? It has not been described. I think elaborating on the seasonal cycle of export and zooplankton could have been interesting.*

R: Yes, the source of export is part of an analysis of export production. We have included in the revised manuscript an additional figure (Fig. 7) and additional text (section 3.5) to elaborate the impacts of the "seasonal paradox" on ecosystem functioning.

L320-332: *This discussion is very speculative and vague. I do not find it very useful.*

R: We have streamlined this part of the discussion after including the results regarding ecosystem functioning as described above.

L340: *Echevin et al. (2020) also investigated the mixed layer evolution under climate change (Figure 7), not only changes in upwelling. I encourage the authors to read the papers they cite more carefully.*

R: We apologise as we did not want to imply that Echevin et al. [2020] did not investigate the mixed layer evolution under climate change. We have rephrased this part to avoid confusion.

L347: *Surface chl only slightly increases in the different simulations (2%-17%, Fig. 12a).*

R: Thank you for pointing this out. We have rephrased this part in the revised manuscript.

L355: *The propagation of the seasonal variability up the food web is not documented in the results sections and only mentioned in the discussion: it is not worth mentioning in the conclusion.*

R: We have added a more detailed presentation of this aspect to the results and discussion sections.

L356: *what are the remaining open questions about the interactions behind the mixed layer and upwelling dynamics? Be more specific.*

R: We conclude that more research is required on how physics will affect ecology, in particular how the mixed layer and upwelling jointly affect food web processes and thus the ecosystem.

Reply to reviewer #2

We would like to thank you for your valuable feedback and your supportive and constructive comments. Your review has helped to improve the clarity of our manuscript. In reply to your comments, we have put additional effort into clarifying the methodology and note where results might be particularly model dependent. Also, we have extended our discussion with regard to climate change. Please find our point-by-point responses below:

L58-60 : *QuickSCAT only cover the periods 1999-2008 so which wind forcing is used to cover the period 1990-2010? please clarify if this in an hindcast run or a climatological simulation.*

R: In the revised manuscript, we have clarified that this study is based on a climatological simulation (L52).

L70-73: *please indicate that the BioEBUS model was first used to simulate the Peru biogeochemistry by Montes et al. (2014) Montes L, B. Dewitte, E. Gutknecht, A. Paulmier , L Dadou, A. Oschlies and V. Garçon, 2014: High-resolution modeling the Oxygen Minimum Zone of the Eastern Tropical Pacific: Sensitivity to the tropical oceanic circulation. J. Geophys. Res.-Oceans. 119, doi:10.1002/2014JC009858*

R: Thank you for pointing us to the first study that employed BioEBUS in the Peruvian system. We have included the reference in the Methods section when introducing BioEBUS.

L78-79 : *"In this study, we use monthly output of the final five years for our analyses (years 26-30)" It is not clear with which forcing the spin-up is done and if is this is a repetitive selected year. Earlier it is mention that the simulation covers the period 1990-2010 so to which actual years correspond years 26-30?*

R: We apologise for not having been more clear about the model set-up. We used the same climatological forcing for the spin-up and output years. No actual years correspond to years 26-30 of the simulation. We have clarified this in the manuscript (L87).

Section 2.2.: *It is not clear if all the terms associated to BIO are calculated on-line or off-line. If this is off-line, the use of monthly-mean mixed-layer depth for the vertical integration could yield errors that would be worth estimating. Mixed-layer depth can vary sharply at high-frequencies.*

R: We have clarified in the revised manuscript that the fluxes associated with BIO were saved as output from the model for each grid box as monthly averages and later were (offline) integrated over the MLD using the croco-tools (L97-102). Indeed, the offline integration of the fluxes over the MLD may introduce a bias due to fluctuations within a month. To estimate the differences of the on- and offline calculations, we compared the model output of the monthly mean of the online integration of the tendency of the phytoplankton biomass with the offline integration (Fig. R2.1). The on- and offline calculations match fairly well, with a slight underestimation of the tendency of phytoplankton biomass if the term is calculated offline.

Figure 1: *"Spatial distribution of the seasonality of surface chlorophyll" what is seasonality exactly? Amplitude of the annual cycle? Seasonality should be defined somewhere.*

R: Seasonality here represents the amplitude of the annual cycle. We have clarified this in the revised manuscript (L140) and figure caption.

L126, Figure 2: *The region for averaging the data for the other EBUS is not defined (?). It should be indicated in the text of the caption for clarity. Please also indicate the results for the Chile EBUS.*

R: We picked the regions based on Chavez and Messié [2009]. We have added Fig. A2 in the appendix that illustrates the regions we used to average over.

Our study is conceptual and meant to focus on processes rather than comparing regional details. Testa et al. [2018] finds that the paradoxical correlation between upwelling and surface chlorophyll is not observed in the Chile EBUS, with low surface chlorophyll in austral winter when MLD is deep and upwelling is weak. The seasonality observed off Chile might resemble the Californian system but it would require a more in-depth analysis to provide conclusions. Therefore we here follow Chavez and Messié [2009] and do not include the Chile EBUS.

Caption of Figure 2: *"upwelling (estimated based on winds from QuikSCAT, in Sv" Do you mean from Ekman transport or Ekman pumping, or from both?. Please provide details on how upwelling intensity is calculated.*

R: The upwelling values (estimated based on winds from QuikSCAT, in Sv) are digitized from Chavez and Messié [2009]. The 'upwelling' is calculated as a combination of Ekman transport and Ekman pumping [Messié et al., 2009, section 3.2 therein].

L154-156: *"In other words, more nutrients only have a strong local positive effect if concentrations are low / would be low otherwise." This sentence is not clear; please rephrase*

R: We have rephrased the sentence in the revised manuscript (L167-168).

L163-165: *"In the model, surface chlorophyll and nitrogen concentrations together with upwelling intensity and MLD all display a 40-60% seasonal variability" what is it meant by "40-60% seasonal variability"? Please clarify and rephrase.*

R: Here, "40 - 60% seasonal variability" refers to the amplitude of the annual cycle relative to the annual mean. We have rephrased the sentence in the revised manuscript (L173-175).

L194-195: *"We separate biological processes (e.g. primary production, grazing from zooplankton, natural mortality, exudation, sinking) and physical processes (mixing, advection and entrainment) that affect the integrated biomass (Fig. 4b)." The detailed equation should be provided along with details on the method for integrating vertically within a seasonally varying mixed-layer. How do you calculate entrainment for instance?*

R: Thanks for pointing out that we missed to explain how we calculated the entrainment. We updated the full equation of the MLD budget of phytoplankton biomass (Equ. 1) and added an explanation how we calculate entrainment to the revised manuscript (L97-102).

L197-198: *"Most biological and physical processes decrease from the start (t1) to the end (t2) of the decline phase (Fig. 4cd)." Biological processes should balance physical processes so when the*

*former increase the later should decrease? We understand from figure 4b that physical processes were multiplied by -1? Could you please clarify and provide details in the text of the caption.*

R: We have added detail to Equ. 1 in the manuscript, defining each of the fluxes. In addition, we have split up Fig. 4 into two figures (now Fig. 4-5) to make the section easier to understand, and added a more detailed explanation to the manuscript (section 3.4.1).

L272-273: *"As we just argued in the previous paragraphs using the differences of the seasonalities of MLD and upwelling in the Peruvian." Connect this sentence to the next one?*

R: Corrected (L309-311).

*The discussion on the impact of global warming is a bit frustrating since it is only based on the implication of a reduced mixed-layer depth in the future. It could be extended to the expected changes in the tendency terms discussed in the paper.*

R: We agree that extending the expected changes to other tendency terms we discussed in the paper would be nice. We have expanded the impact of global warming in the last paragraph of the section conclusions and potential implications (L379-392).

*Figure C4: "The correlation coefficient (R2=0.81) is shown for the decline phase" the correlation uses only 5 points so it is certainly associated to a low level of confidence?*

R: In this case, we specifically selected the five points of the decline phase to address the role of advection during the phase that constitutes the seasonal paradox. As evident from Fig. C4, the intensity of upwelling appears not to be the key driver of phytoplankton advection throughout the year (the magnitude of the concentration of phytoplankton biomass in the MLD plays a role as well). However, there is a rather strong correlation from April to August during the decline phase that we focus on. In summary, we would like to highlight that upwelling and advection are correlated specifically in the decline phase, which is why we used only these five months for the correlation. To avoid confusion in comparison to correlations that we show in other figures, we have marked in the manuscript and noted in the figure captions what time periods we used in each of the figures to calculate the correlations over.

---

## Author Response (AR2)

I find that the authors have made a substantial effort to take into account my comments. I think the paper could be published after the minor revisions I listed below.

L176: *open squares*

R: Corrected.

Figure 5: *These budgets can be difficult to interpret. I think it would be helpful to explain why there is less grazing during this phase. Besides, it is not clear to me why vertical mixing contributes positively as the mixed layer increases: what is your interpretation of this process? I would not comment on entrainment and add this weak term to vertical mixing.*

R: In an attempt to improve the clarity of argumentation, we have modified Fig. 5 such that the colours of the bars (indicating changes between t1 and t2) now match the colours of the time series.

Less grazing during this phase could be due to a series of processes, including the amount or composition of food that is available to zooplankton for grazing. As grazing is not contributing to the seasonal paradox, we prefer not to go into detail on grazing but instead focus on the budget terms that do contribute to the seasonal paradox.

Regarding mixing, throughout the year, mixing continuously mixes phytoplankton out of mixed layer, as phytoplankton concentration above the mixed layer depth is consistently higher than the concentration below the mixed layer. However, the vertical gradient across the base of the mixed layer is getting smaller from t1 to t2, as the "above the base of mixed layer" phytoplankton concentration is decreasing more dramatically as the mixed layer is deepening compared to "below the base of mixed layer". Therefore, though "mixing" is always negatively contributing to biomass accumulation, the absolute value of "mixing" is decreasing from t1 to t2.

We agree with your suggestion to combine entrainment and mixing, and have done so in the revised manuscript.

L235: *refer explicitly to the figure in the text please.*

R: Done.

L236: *please explain how you obtain 60% (and 40% for temperature) using equation 3.*

R: The contribution is calculated based on the multiplicative function of the light-, temperature-, and nitrogen-related growth factors. We estimated based

on the product rule and the change of each single growth factor over the decline phase how much each factor would contribute to the total growth factor change. To make this clearer, we have added in L237:
"Estimated from the product rule for differentiation and the multiplicative relation of growth factors shown in Eq. 2 & 3".

L255: *The minor role of the seasonality of radiation was also shown in Echevin et al 2008.*

R: Thanks for pointing this out. We have added that the results agree with findings by Echevin et al 2008.

L257: Please refer to figure 5c explicitly in the text.

R: Done.

Section 3.5: Figure B5 is not mentioned anywhere in the text, which is a shame as the authors made the effort to compare with observations. Besides, it is really difficult to evaluate the model's skill to simulate zooplankton biomass from Figure B5. A scatterplot and a correlation value would give a clearer more quantitative view.

R: Thank you for this suggestion. As suggested by the reviewer, we have added the scatter plot (Fig. B5c) and have updated the manuscript accordingly. While the spread is large, the dots mostly scatter around the 1:1 line. The model is not able to capture the observed instances of very high zooplankton biomass, though these are outliers where also the observational data quality may be an issue. Observational mesozooplankton data have a very large error bar because of differences in sampling methods, mesh sizes, seasonal sampling bias, and diurnal vertical migration (O'Brien, 2007). This is why we expect simulated zooplankton biomass not to match observational estimates as well as physical variables do.

L279: why "must" export overcompensate? Please rephrase, the entire sentence is unclear to me.

R: Export efficiency is calculated as the ratio of export to primary production. For export efficiency showing the same seasonality as export (numerator) and primary production (denominator), namely high values in summer and low values in winter, the term in the numerator (export) has to show a larger seasonal variation then the denominator (primary production).

L294: *phytoplankton was diluted*

R: Done.

L335: Messié and Chavez (2015) also mention the effect of nearshore eddies, which transport upwelled nutrients offshore and downward. This effect has been evidenced in several works (e.g. Gruber et al., 2011, Lathuilière et al., 2010) and should be discussed in section 4.2.

R: We added the sentence (L335-336):
"Eddies have been found to favour offshore export and subduction of phytoplankton and nutrients (Lathuiliere et al.,2010; Gruber et al., 2011; Messie and Chavez, 2015)."

L347: Mesozooplankton is . . .

R: Done.

L380: "we argue...": Could you explain more clearly what are the open questions about the interactions behind the mixed layer and upwelling dynamics and food web processes? I find the last sentences of this paragraph very vague and I do not see how they will motivate future studies.

R: We rephrased L380-381 and specified:
Given that changes in MLD are correlated to many ecosystem components related to plankton ecosystem functioning, we argue for a more thorough understanding of the impact of the seasonal paradox on the ecosystem. In particular effects on the trophic transfer of energy through the plankton food web to higher trophic levels such as fish will determine ecosystem functions like trophic transfer efficiency, fish production, and ultimately potentially fisheries yields.

Legend of Figure B2: *it would be nice to cite a publication to have details on the two cruises.*

R: Thank you for pointing out the missing reference. We have added the reference provided at PANGAEA with the data set, Thomsen et al., 2016.

L440 and Figure B3: It would be useful to explain how the model mixed layer was computed and whether the method differs from the one used in the ARGO and De Boyer Montegut data sets. Please explain also how you obtained the error bars. The model mixed layer tends to be shallower than ARGO data in june, july, august, september, november. ARGO is also much shallower in Boyer Montegut data set. This need to be commented. You may also be interested to cite a very recent paper which shows that the mixed layer depth bias can reduced when the chlorophyll shading effect is taken into account (Echevin et al., 2021).

R: We have added in the text how we computed the mixed layer depth. To be consistent with the observational data, we calculated the mixed layer depth

offline as the depth with a 0.2°C temperature difference to the surface. We are also more clear in the legend now that the error bar indicates the standard deviation of all data points in the focus area in the respective month. A more detailed comparison between ARGO data, the de Boyer Montégut data, and the simulated mixed layer depth is added in the text. We also included the Echevin et al., 2021 paper, with the argument that the chlorophyll shading effect may partially explain the deep bias of the modelled MLD.

Figure B4: to be consistent with the previous figure error bars (standard deviation) should be shown for MODIS and CROCO.

R: Figure B4 has been updated accordingly.

L451: *It is nice of the authors to have made the effort of comparing zooplankton data with their model results. However I am not very convinced by the comparison. The large scale meridional gradient (low zoo in the south, more zoo in the north) seems to be roughly reproduced, but the large-scale zonal gradient and cross-shore gradient near the coast do not seem well reproduced to me. There tends to be a higher zooplankton biomass to the north west of the white box. A model vs data scatterplot would be also useful. Are the data annual averages? Does the seasonality of zoo in the box fit with the model's seasonality?*

R: Next to adding a scatterplot of observed versus modelled zooplankton biomass as suggested by the reviewer in a comment above, we now show the spatial pattern of modelled and observed zooplankton in two separate maps instead of one to highlight the large-scale pattern that we feel is well represented by the model. We have added also a more detailed discussion of the spatial pattern, and the agreements and disagreements of the modelled zooplankton biomass versus observed estimates, such as the offshore high bias of the model. We suggest that the offshore zooplankton high bias is likely related to the high bias of offshore phytoplankton biomass, which in turn presumably results from a lack of iron limitation in the model. Furthermore, the observational estimates of zooplankton biomass are based on a wide range of methods and accordingly have a large uncertainty that is difficult to quantify. An agreement of model and observations in magnitude and large scale pattern is therefore a meaningful result.

The observational estimates are temporally-averaged because they are very sparse, and there is a summer bias in the observational sampling, with the observational data only covering half of the year. We do call for more extensive observations that cover in particular the full seasonal cycle, we have added this latter point to the manuscript.